# Enhancing Deep Graph Neural Networks via Improving Signal Propagation

## Abstract

Graph neural networks (GNNs) suffer from the *curse of depth*, a phenomenon where performance degrades significantly as network depth increases. In this work, we aim to provide a more principled analysis and solution via the lens of signal propagation. We identify three metrics for a good signal propagation in graph neural nets: forward-propagation, backward-propagation, and graph embedding variation (GEV). We prove that traditional initialization methods, which deteriorate the performance of deep GNNs, fail to simultaneously control the three metrics. To tackle this issue, we develop a new GNN initialization method called **S**ignal **P**ropagation **o**n **G**raph (SPoGInit), which searches for weight variances that minimize the three metrics. In various datasets, SPoGInit achieves notable performance enhancements in node classification tasks as GNNs grow deeper. For instance, we observed a 2.2% gain in test accuracy on OGBN-Arxiv dataset as the depth increases from 4 to 64.

## 1 Introduction

Increasing depth has been an important theme in the development of neural networks. For instance, from AlexNet (Krizhevsky et al., 2012), VGG19 (Simonyan & Zisserman, 2015) to ResNet He et al. (2016), the depth of CNN (convolutional neural network) has increased from 8, 19 to 52, and the corresponding test accuracy on ImageNet has increased from 63.3%, 74.4% to 78.57%. The theoretical benefit of depth is often considered to be strong representation power: it was proven that to represent a deep network with a small width by a shallow network, exponentially many neurons are required (Telgarsky, 2015; Eldan & Shamir, 2016; Liang & Srikant, 2017).

Nevertheless, unleashing the power of deep nets requires extra techniques to handling the training difficulties. For instance, He et al. (2016) pointed out that at the time of writing their paper, increasing the depth beyond 20 will lead to worse performance, and they proposed a new technique (skip connection) to train 50+ layer CNN. Designing initialization is another method to handle the training issues of deep nets Xiao et al. (2018).

In the realm of Graph Convolutional Networks (GCNs), it is natural to explore deeper architectures. Deeper GCNs inherently possess stronger representation power, and thus hold potential advantages in handling complex graph data, including those that display long-range dependencies among nodes. Nonetheless, empirical studies indicate that increasing the depth of a GCN often deteriorate performance, rather than enhance it. This phenomenon, which we refer to as the *curse of depth*, presents a substantial challenge in the development of effective GCNs. In recent years, over-smoothing (Li et al., 2018; Oono & Suzuki, 2019) has been identified as one of the major reasons for the curse of depth. Over-smoothing occurs when, as a GCN becomes deeper, embeddings among different nodes become increasingly similar, rendering nodes challenging to differentiate. A variety of approaches have been explored to tackle the over-smoothing issue within the GCN family, such as nodes or edges dropping techniques (Srivastava et al., 2014; Zou et al., 2019; Rong et al., 2020; Huang et al., 2020; Lu et al., 2021), normalization techniques (Ioffe & Szegedy, 2015; Zhao & Akoglu, 2020; Zhou et al., 2020; Yang et al., 2020; Zhou et al., 2021b; Li et al., 2020; Guo et al., 2023), and regularization techniques (Chen et al., 2020a; Yang et al., 2020; Zhou et al., 2021a). Despite their good performance, they have not fully alleviated the curse of depth. In fact, the optimal performance for GCNs in most of these studies is still achieved with less than 20 layers, suggesting that the curse of depth continues to constrain the potential of GCN. See more detailed discussion of their optimal performance and the

corresponding depths in Appendix F. Such limitation indicates an ongoing need for new perspectives and strategies to resolve the curse of depth.

Inspired by signal propagation (SP) theory in deep neural networks (DNNs) (Poole et al., 2016; Schoenholz et al., 2017; Pennington et al., 2017; 2018; Hanin, 2018), we realize that over-smoothing is also one type of signal propagation issue, though it is only present in GCNs and occurs due to the graph structure. This understanding has two implications. First, it is an SP issue, and thus may be addressed via the common SP-handling methods, such as initialization design. Second, it is just ONE SP issue, and other SP issues exist, thus addressing just over-smoothing may not be enough to address the curse of depth.

Based on this understanding, we propose the following framework to address the curse of depth in GCNs: (i) consider both the graph-dependent SP issue (oversmoothing) and generic SP issue (forward signal propagation and backward signal propagation); (ii) design a proper initialization to resolve these issues simultaneously. More specifically, we consider three metrics to assess the quality of GCN initialization: forward signal propagation (FSP) and backward signal propagation (BSP), and graph embedding variation (GEV). Here the GEV metric (an oversmoothing metric) is graph dependent, and the FSP and BSP metrics are generic. With these metrics, we provide theoretical analysis of classical initiazation and propose a new intialization scheme, as detailed below.

- **Theory: Analysis of Classical Initialization**: We prove that traditional initialization methods for both vanilla GCNs and residual GCNs (ResGCNs) do not control all three metrics simultatenously. We provide experiments to show that the traditional initialization methods can indeed cause the explosion/vanishing of one or more metrics. These results help explain why traditional initialization methods cannot resolve the curse of depth issue in GCNs.
- **Algorithm: SPoGInit**: We propose a new initialization design method called Signal Propagation on Graph (SPoGInit), which is to use an optimization algorithm to find weight variances to control all three signal propagation (SP) metrics. For ResGCNs, a special version of this initialization, termed SPoG-ResGCN, is introduced. Experiments show that on complex graph data, with the proposed initialization methods the performance increases as the depth increases, while other initialization methods or models display deteriorating performance as the depth increases.

## 2 PRELIMINARIES AND BACKGROUND

For any integer $N \in \mathbb{N}$, we define $[N] := \{1, 2, \ldots, N\}$. For brevity, we use $\theta$ to denote the collection of trainable parameters in a GCN model. For additional useful notation, see Appendix B.1.

### 2.1 GRAPH CONVOLUTIONAL NETWORKS

**Featured graph.** Let $\mathcal{G} = (\mathcal{V}, \mathcal{E})$ be an undirected graph, where $\mathcal{V}$ is the set of nodes with $|\mathcal{V}| = n$, and $\mathcal{E}$ is the collection of edges. Assume that each node is associated with a $d_0$-dimensional feature and a label belonging to $[C]$. Let $x_i \in \mathbb{R}^{d_0 \times 1}$ and $y_i \in [C]$ denote the feature and the label of node $i$, respectively. Define the node feature matrix as $X = (x_1^T, x_2^T, \ldots, x_n^T)^T \in \mathbb{R}^{n \times d_0}$. Let $A = (\mathbb{1}_{\{(i,j) \in \mathcal{E}\}})_{i,j \in [n]} \in \mathbb{R}^{n \times n}$ represent the adjacency matrix and $D = \text{diag}(A\mathbf{1}_n) \in \mathbb{R}^{n \times n}$ represent the degree matrix. Further, $\tilde{A} = A + I$ and $\tilde{D} = D + I$ denote the adjacency matrix degree matrix of graph $\mathcal{G}$ with self-loop added to each node. Finally, the normalized adjacency matrix is given by $\hat{A} = \tilde{D}^{-\frac{1}{2}} \tilde{A} \tilde{D}^{-\frac{1}{2}}$.

**Vanilla GCN.** Vanilla GCN (Kipf & Welling, 2017) stacks neighborhood aggregations and feature transformations alternately. Specifically, let $H^{(l)}, X^{(l)} \in \mathbb{R}^{n \times d_l}$ denote the pre-activation and the post-activation embedding matrix at the $l$-th layer of the vanilla GCN, respectively. They are defined recursively by
$$H^{(l)} := \hat{A} X^{(l-1)} W^{(l)} + \mathbf{1}_n \cdot b^{(l)}, \quad X^{(l)} := \sigma(H^{(l)}),$$
where $W^{(l)} \in \mathbb{R}^{d_{l-1} \times d_l}$ and $b^{(l)} \in \mathbb{R}^{1 \times d_l}$ are the weight and the bias term at the $l$-th layer, respectively. The input to the first layer is given by $X^{(0)} = X$, and the output matrix of an $L$-layer vanilla GCN is $H^{(L)} \in \mathbb{R}^{n \times C}$, which is then fed into a softmax layer to obtain the predicted labels.

**ResGCN.** Inspired by He et al. (2016), ResGCN (Kipf & Welling, 2017) combines residual connections with vanilla GCN. An $L$-layer ResGCN adds skip connections to the post-activation

embeddings, i.e.,

$$H^{(l)} := \hat{A}X^{(l-1)}W^{(l)} + \mathbf{1}_n \cdot b^{(l)}, \quad X^{(l)} := \alpha\sigma(H^{(l)}) + \beta X^{(l-1)}, \quad \forall l \in [L],$$

where $\alpha, \beta \in \mathbb{R}$ are gating hyper-parameters.[1] Linear transformations (trainable) are applied before $X^{(0)}$ and after $H^{(L)}$ to ensure consistency of embedding sizes.

## 2.2 Initialization

We consider the following class of initialization methods. At initialization, all $W_{k'k}^{(l)}$ are i.i.d. and satisfy $\mathbb{E}[W_{k'k}^{(l)}] = 0$, $\mathrm{Var}[W_{k'k}^{(l)}] = \sigma_w^2/d_{l-1}$; all $b_k^{(l)}$ are initialized to be 0 for any $k' \in [d_{l-1}], k \in [d_l], l \in [L]$.

Two widely used random initialization methods, LeCun initialization and Kaiming initialization (He et al., 2015) fit into this framework with $\sigma_w^2 = 1$ and $\sigma_w^2 = 2$ respectively.

- LeCun: $\mathbb{E}[W_{k'k}^{(l)}] = 0$ and $\mathrm{Var}[W_{k'k}^{(l)}] = 1/d_{l-1}$.
- Kaiming (usually for ReLU): $\mathbb{E}[W_{ij}^{(l)}] = 0$ and $\mathrm{Var}[W_{ij}^{(l)}] = 2/d_{l-1}$.

In GCN models, uniform weight distribution with variance $\sigma_w^2 = 1/3$ is also widely used, e.g., in PairNorm (Zhao & Akoglu, 2020), DropEdge (Rong et al., 2020), DropNode (Huang et al., 2020), SkipNode (Lu et al., 2021), GCNII (Chen et al., 2020b). We simply refer to this initialization as "Conventional initialization" in the rest of this paper. Xavier initialization has weight variance $2/(d_{l-1} + d_l) = 1/d$ when hidden layers have the same width $d$.

## 3 Theoretical analysis of GCN initializations

In this section, we evaluate the quality of GCN initializations from three aspects based on the signal propagation (SP) theory as follows.

**Forward signal propagation (FSP)** is responsible to extract abstract and higher-level representations from the input data as the information flows through the network. We propose the *FSP metric* $\mathbf{M}_{\mathrm{FSP}}^{(L)}(\sigma_w^2)$, which is the expected output-input norm ratio $\mathbb{E}_\theta[\|H^{(L)}(\theta)\|_\mathrm{F}^2/\|X\|_\mathrm{F}^2]$. A proper initialization method should prevent $\mathbf{M}_{\mathrm{FSP}}^{(L)}(\sigma_w^2)$ from either vanishing or exploding as $L \to \infty$.

**Backward signal propagation (BSP)** is responsible for updating the weights by utilizing gradients computed via back-propagation. In vanilla GCN, the gradient of $W^{(l)}$ at the $l$-th layer can be decomposed as $\partial\ell/\partial W^{(l)} = \sigma(H^{(l-1)})^T \cdot \hat{A} \cdot [\partial\ell/\partial H^{(l)}]$ where $\ell$ is the training loss. A stable magnitude of $\partial\ell/\partial H^{(l)}$ with respect to the layer $l$ suggests that the gradient is less susceptible to vanishing or exploding. We take $\mathbb{E}_\theta[\|\partial\ell/\partial W^{(1)}\|_\mathrm{F}^2]$ at initialization as the *BSP metric* $\mathbf{M}_{\mathrm{BSP}}^{(L)}(\sigma_w^2)$. A proper initialization method should prevent $\mathbf{M}_{\mathrm{BSP}}^{(L)}(\sigma_w^2)$ from vanishing or exploding as $L \to \infty$.

**Graph embedding variation (GEV) propagation** is responsible for tackling the over-smoothing issue, a GCN-specific problem. A number of existing works (Cai & Wang, 2020; Zhou et al., 2021a) measure over-smoothing severity by Dirichlet energy $\mathrm{Dir}(H^{(L)}) = \sum_{(i,j)\in\mathcal{E}} \|h_i/\sqrt{1+d_i} - h_j/\sqrt{1+d_j}\|^2$, where $h_i$ is the output embedding of node $i$. Dirichlet energy $\mathrm{Dir}(H^{(L)})$ reveals the embedding variation with the weighted node pair distance, and a smaller value of $\mathrm{Dir}(H^{(L)})$ is highly related to the over-smoothing. To eliminate the influence of the embedding norm, we propose the *GEV metric* $\mathbf{M}_{\mathrm{GEV}}^{(L)}(\sigma_w^2)$, which is the expected of normalized Dirichlet energy $\mathbb{E}_\theta[\mathrm{Dir}(H^{(L)})/\|H^{(L)}\|_\mathrm{F}^2]$ at initialization. A proper initialization method should prevent $\mathbf{M}_{\mathrm{GEV}}^{(L)}(\sigma_w^2)$ from vanishing as $L \to \infty$.

## 3.1 Theoretical results for vanilla GCN

We first theoretically evaluate the signal propagation (SP) quality at initialization in vanilla GCN. Due to the nonlinearity and high dimensionality of neural networks, the SP analysis is challenging.

---

[1] The original version of ResGCN (Kipf & Welling, 2017) focuses on the special case $(\alpha, \beta) = (1, 1)$.

In order to simplify it, we study the infinite-width limit of vanilla GCN using mean field theory (Poole et al., 2016; Schoenholz et al., 2017). Different from traditional NNs, GNN blocks involve interactions across nodes, so we have to consider the signal propagation of $n$ nodes as an integrated whole, rather than that of only one data sample in NNs. Under this approximation, all the channels $\{H_{:,k}^{(l)}\}_{k=1}^{d_l}$ of each embedding at the $l$-th layer are i.i.d., following Gaussian distribution $N(\mathbf{0}_n, \Sigma^{(l)})$. The $n \times n$ covariance matrix $\Sigma^{(l)}$ recursively satisfies

$$\Sigma^{(l)} = \sigma_w^2 \hat{A} G(\Sigma^{(l-1)}) \hat{A}, \quad \Sigma^{(1)} = \sigma_w^2 \hat{A} X X^T \hat{A}/d_0,$$

where $G(\Sigma^{(l)}) = \mathbb{E}_{h \sim N(\mathbf{0}_n, \Sigma)}[\sigma(h)\sigma(h)^T] \in \mathbb{R}^{n \times n}$ (see Appendix D.1 for the details). This theoretical framework is referred to as the neural network Gaussian process (NNGP) correspondence. Under the NNGP correspondence, the forward propagation (FSP) metric can be approximated by

$$\mathbf{M}_{\text{FSP}}^{(L)}(\sigma_w^2) \approx \mathbb{E}_{H^{(L)} \sim N(\mathbf{0}_n, \Sigma^{(L)})}\left[\|H^{(L)}\|_{\text{F}}^2 / \|X\|_{\text{F}}^2\right]$$

and the graph embedding variation (GEV) metric can be approximated by

$$\mathbf{M}_{\text{GEV}}^{(L)}(\sigma_w^2) \approx \mathbb{E}_{H^{(L)} \sim N(\mathbf{0}_n, \Sigma^{(L)})}\left[\text{Dir}(H^{(L)})/\|H^{(L)}\|_{\text{F}}^2\right],$$

where $H^{(L)} \sim N(\mathbf{0}_n, \Sigma^{(L)})$ means all columns (channels) of $H^{(L)} \in \mathbb{R}^{n \times C}$ are i.i.d. $N(\mathbf{0}_n, \Sigma^{(L)})$.

Now we analyze the signal propagation of GCN under various activation functions. We start with ReLU since it is the most commonly used activation in popular GCN models (e.g., (Zhao & Akoglu, 2020; Rong et al., 2020; Huang et al., 2020; Lu et al., 2021; Chen et al., 2020b)). The following theorem states that under ReLU activation, if the initial weight variance $\sigma_w^2 \leq 2$, which covers Conventional, Kaiming, and LeCun initialization, deep vanilla GCNs suffer from poor FSP and GEV.

**Theorem 3.1.** *Under the NNGP correspondence approximation, when the activation function $\sigma$ is ReLU, we have*

1. *If $\sigma_w^2 = 2$, either the limit graph embedding variation metric $\lim_{L \to \infty} \mathbf{M}_{GEV}^{(L)}(\sigma_w^2) = 0$ or the limit forward-propagation metric $\lim_{L \to \infty} \mathbf{M}_{FSP}^{(L)}(\sigma_w^2) = 0$;*

2. *When $\sigma_w^2 < 2$, the forward-propagation metric $\mathbf{M}_{FSP}^{(L)}(\sigma_w^2) \leq \frac{2C}{d_0} \cdot (\sigma_w^2/2)^L$ for any $L \geq 1$.*

Part 1 of Theorem 3.1 shows that under Kaiming initialization in ReLU-activated vanilla GCN, either $\mathbf{M}_{\text{FSP}}^{(L)}$ or $\mathbf{M}_{\text{GEV}}^{(L)}$ vanishes as $L \to \infty$. Part 2 of Theorem 3.1 characterizes the shrinkage of $\mathbf{M}_{\text{FSP}}^{(L)}$ when $\sigma_w^2$ is even less than that of Kaiming initialization.

**Theorem 3.2.** *Under the NNGP correspondence approximation, when the activation is ReLU, the graph embedding variation metric $\mathbf{M}_{GEV}^{(L)}$ is independent of $\sigma_w^2$.*

Theorem 3.2 states that it is impossible to improve the graph embedding variation $\mathbf{M}_{\text{GEV}}^{(L)}(\sigma_w^2)$ by simply refining $\sigma_w^2$ for ReLU-activated vanilla GCN. In other words, the over-smoothing issue cannot be resolved by adjusting weight variance $\sigma_w^2$ in ReLU-activated vanilla GCN.

We now provide numerical evidence for Theorem 3.1 and 3.2. The purple lines in Figure 1(a)-(c) illustrate the shrinkage of the three SP metrics under Conventional initialization as the network depth $L$ increases. Figure 1(a) when $\sigma_w^2$ presents the vanishing pattern of $\mathbf{M}_{\text{FSP}}^{(L)}(\sigma_w^2)$ is no greater than that of Kaiming initialization, which validates Theorem 3.1. Figure 1(b) shows that $\mathbf{M}_{\text{BSP}}^{(L)}(\sigma_w^2)$ transits from vanishing to stable, and then to exploding as $\sigma_w^2$ increases. Figure 1(c) shows that $\mathbf{M}_{\text{GEV}}^{(L)}(\sigma_w^2)$ cannot be improved via merely changing $\sigma_w^2$, which validate Theorem 3.2.[2]

Different from ReLU-activated GCNs, Figure 1(f) shows propagation transits from vanishing to stable for tanh-activated models as $\sigma_w^2$ increases. With proper $\sigma_w^2$, stable propagation for all three types of signals can be achieved; see the orange lines in Figure 1(d)-1(f). A theoretical result of the forward propagation for tanh-activated vanilla GCNs is provided in Appendix D.5.

---

[2]In all the figures illustrating signal propagation metrics, disappearing nodes and vertical lines are caused by surpassing the machine precision. Specifically, the vanishing forward propagation metric result in vertical lines in the plots of the graph embedding variation metric, while the exploding forward propagation metric leads to node disappearance in the plots of the graph embedding variation metric.

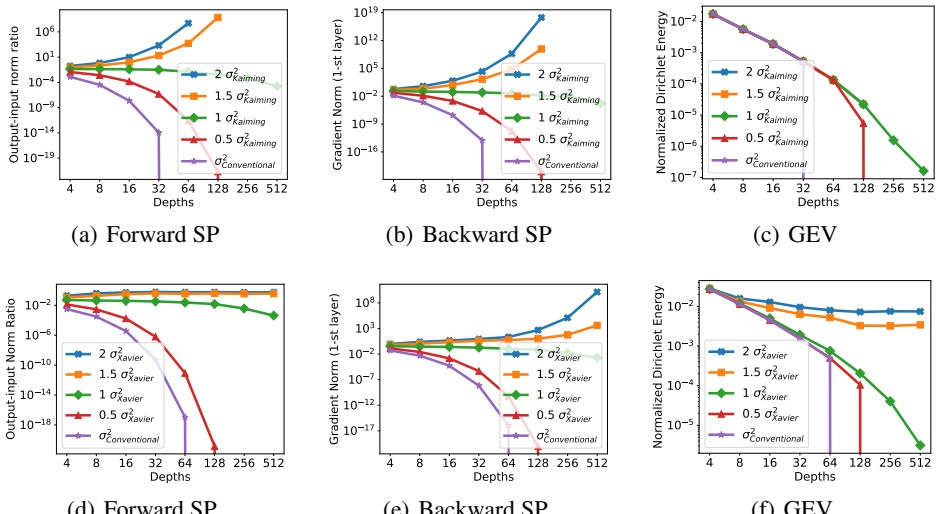

Figure 1: Plots of (a,d) forward metrics, (b,e) backward metrics, and (c,f) graph embedding variation metrics of deep vanilla GCNs with different initialization variances and activations on Cora. (Sub-figures (a)-(c) are for ReLU activation, while sub-figures (d)-(f) are for tanh activation.) The choice of initialization variance plays a crucial role in forward and backward propagation. The graph embedding variation propagation can be made stable with proper initialization variance for tanh activation, but not for ReLU activation.

## 3.2 THEORETICAL RESULTS FOR RESGCN

Similarly to vanilla GCN, the curse of depth has also been reported in deep ResGCN (Huang et al., 2020; Rusch et al., 2023a). In this subsection, we focus on the signal propagation in ResGCN.

For simplicity, we study linear ResGCN with identity activation in the theoretical analysis. Such a simplification is very common in NN theory (Saxe et al., 2014; Xu et al., 2021). Similar to vanilla GCN, all the channels of $H^{(L)}$ are i.i.d. $N(\mathbf{0}_n, \tilde{\Sigma}^{(L)})$ under the infinite-width limit (a.k.a. NNGP correspondence). The $n \times n$ covariance matrix $\tilde{\Sigma}^{(l)}$ recursively satisfies

$$\tilde{\Sigma}^{(l)} = \sigma_w^2 \hat{A} \tilde{\Sigma}^{(l-1)} \hat{A} + \tilde{\Sigma}^{(l-1)}, \quad \tilde{\Sigma}^{(1)} = \sigma_w^4 \hat{A} X X^T \hat{A}/d_0, \tag{1}$$

See Appendix E.1 for the details.

The following theorem implies that linear ResGCN may suffer from forward signal explosion and over-smoothing under the NNGP approximation at initialization.

**Theorem 3.3.** *Suppose that there exists an eigenvector $u$ of $\hat{A}$ corresponding to the eigenvalue $1$, such that the input feature $X \in \mathbb{R}^{n \times d_0}$ satisfies $X^T u \neq \mathbf{0}_{d_0 \times 1}$. Under the NNGP correspondence for linear ResGCN, if $\alpha^2 \sigma_w^2 + \beta^2 > 1$ and $\alpha \neq 0$, then we have*

$$\lim_{L \to \infty} \mathbf{M}_{FSP}^{(L)}(\sigma_w^2) = \infty \quad and \quad \lim_{L \to \infty} \mathbf{M}_{GEV}^{(L)}(\sigma_w^2) = 0.$$

Since $(\alpha, \beta) = (1, 1)$ for the original ResGCN (Kipf & Welling, 2017), $\alpha^2 \sigma_w^2 + \beta^2 > 1$ and $\alpha \neq 0$ always hold for any *nonzero* initialization variance, which indicates exploding $\mathbf{M}_{\text{FSP}}^{(L)}(\sigma_w^2)$ and shrinking $\mathbf{M}_{\text{BSP}}^{(L)}(\sigma_w^2)$.

Numerical experiments demonstrate that the consequences of Theorem 3.3 can be observed on ResGCNs with non-linear activations. In Figure 2, we plot the FSP and the GEV of ReLU-activated ResGCN with different initialization variances. We see that the widely used Conventional and Kaiming initialization schemes (Huang et al., 2020; Kipf & Welling, 2017) (and essentially any non-zero initialization variance) lead to exploding forward propagation and over-smoothing.

In summary, the discussions in Sections 3.1 and 3.2 provide a theoretical guarantee that the traditional initialization schemes utilized in both vanilla GCN and ResGCN fail to achieve proper SP. To address this challenge, we will introduce new initialization schemes in the subsequent section.

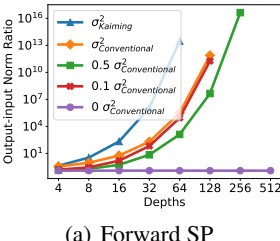
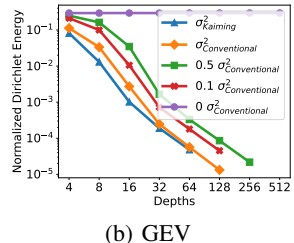

(a) Forward SP

(b) GEV

Figure 2: (a) The forward metrics and (b) the graph embedding variation metrics of ReLU-activated deep ResGCN on Cora. ResGCNs with non-zero initialization variances always suffer from exploding forward propagation and over-smoothing.

## 4 SPoGInit: INITIALIZATION GUIDED BY SIGNAL PROPAGATION ON GRAPH

Section 3 theoretically shows that conventional initialization approaches fail to ensure effective signal propagation for GCNs. In essence, finding a single initialization variance that caters to every layer, while meeting all three SP metrics, proves to be a daunting task. One promising approach is to permit distinct initialization variances across different layers. This strategy allows various layers to synergistically complement each other, leading to more efficient signal propagation.

However, allowing distinct layer-wise variances introduces a new challenge. Specifically, it is not straightforward to design a unified criterion to set the variances for GCNs with varying depths. To resolve this challenge, we introduce the SPoGInit (Signal Propagation on Graph guided Initialization) method. For any GCN (with a given depth), SPoGInit formulates and solves an optimization problem, so as to determine the layer-wise initialization variances that meet the SP requirements.

Given a GCN with $L$ layers, we denote the variance of the $l$-th layer by $\sigma_{w,l}^2$. SPoGInit solves the following optimization problem

$$\underset{\{\sigma_{w,l}\}_{l=1}^L}{\text{minimize}} \quad w_1 \mathbf{V}_{\text{FSP}} + w_2 \mathbf{V}_{\text{BSP}} - w_3 \mathbf{M}_{\text{GEV}}^{(L)} \tag{2}$$

where $\mathbf{V}_{\text{FSP}} := (\mathbf{M}_{\text{FSP}}^{(1)}/\mathbf{M}_{\text{FSP}}^{(L-1)} - 1)^2$ encourages consistent forward propagation metrics across hidden layers, while $\mathbf{V}_{\text{BSP}} := (\mathbf{M}_{\text{BSP}}^{(2)}/\mathbf{M}_{\text{BSP}}^{(L-1)} - 1)^2$ targets consistent backward propagation metrics, with the numbers in parentheses indicating the layer index corresponding to the gradient norm. Besides, $w_1, w_2, w_3 > 0$ are pre-defined for balancing these three SP metrics. During the implementation of SPoGInit, we adjust the weight initialization variances across layers by gradient descent algorithm. More details about SPoGInit are in Appendix G.

### 4.1 SPoGInit FOR ResGCN

SpoGInit can be applied to both vanilla GCN and ResGCN. While applying SPoGInit for Res-GCNs, we observed an interesting phenomenon that the average magnitude of residual blocks $\alpha\sigma(\hat{A}X^{(l-1)}W^{(l)})$ converges to 0 in Figure 3(a). It implies that in order to achieve satisfactory SP in ResGCNs, the "signal" of the first layer should be preserved in all the following layers. That is, we should ensure each hidden embedding $H^{(l)}$ to be identical to $H^{(1)}$.

The above requirement leads to three potential initialization designs for ResGCNs regarding the initialization variances and the gating parameters:

**(D1).** $\alpha^{(l)} = 0, \sigma_{w,l}^2 = 0$; **(D2).** $\alpha^{(l)} = 0, \sigma_{w,l}^2 > 0$; **(D3).** $\alpha^{(l)} > 0, \sigma_{w,l}^2 = 0$.

All these three designs set $\beta^{(l)} = 1$. Here, $\alpha^{(l)}, \beta^{(l)}$ denote the corresponding gating parameters at the $l$-th layer, which are set to be trainable parameters to preserve the expressive power of ResGCNs.

We test the performance of the three designs. Figure 3(b) demonstrate that **(D2)** significantly outperforms **(D1)** and **(D3)**. Consequently, we choose **(D3)** as the initialization strategy called as **SPoG-ResGCN** for ResGCN. SPoG-ResGCN's corresponding network is

$$H^{(l)} := \hat{A}X^{(l-1)}W^{(l)} + \mathbf{1}_n \cdot b^{(l)}, \quad X^{(l)} := \alpha^{(l)}\sigma(H^{(l)}) + \beta^{(l)}X^{(l-1)}, \quad \forall l \in [L],$$

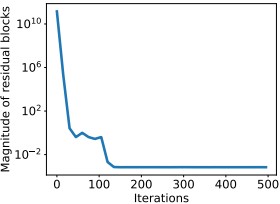 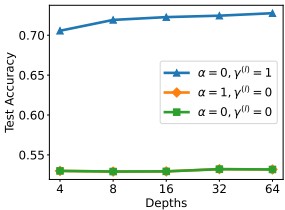

Figure 3: (a) The average magnitude of residual blocks in ResGCN during the SPoGInit on the Cora dataset. (b) the average test accuracies of ResGCN with different initialization designs. We find that the average magnitude of residual blocks rapidly reaches close to 0 during SPoGInit searching and the initialization design $\alpha = 0$, $\beta = 1$, $\sigma_{w,l}^2 > 0$ significantly outperforms other designs.

where the trainable parameters $(\alpha^{(l)}, \beta^{(l)}) = (0, 1)$ and $\text{Var}[W_{ij}^{(l)}] = \sigma_w^2/d$ at initialization.[3]

From the theoretical perspective, SPoG-ResGCN can also be deducted from Theorem 3.3. Since $\alpha^2 \sigma_w^2 + \beta^2 > 1$ and $\alpha \neq 0$ leads to exploding forward metric $\mathbf{M}_{\text{FSP}}^{(L)}$ and vanishing graph embedding variation $\mathbf{M}_{\text{GEV}}^{(L)}$, a straightforward way to resolve it is to set $\alpha^{(l)} = 0$. If so, each layer of ResGCN becomes $X^{(l)} = \beta^{(l)} X^{(l-1)}$. Again, to preserve forward SP, a natural way is to set $\beta^{(l)} = 1$.

In the experimental section, we demonstrate how SPoGInit, when applied to both vanilla GCN and ResGCN, effectively improves signal propagation and alleviates the performance decline in deep GCNs.

## 5    EXPERIMENTS

In this section, we demonstrate the advantage of the proposed SPoGInit in training deep graph neural nets. Due to limited space, details of the datasets, experimental settings, and hyperparameters are given in Appendix H.1 and H.2.

### 5.1    EXPERIMENTS FOR VANILLA GCNs

We first examine **whether SPoGInit tackles the signal propagation and performance degradation of deep GCNs**.

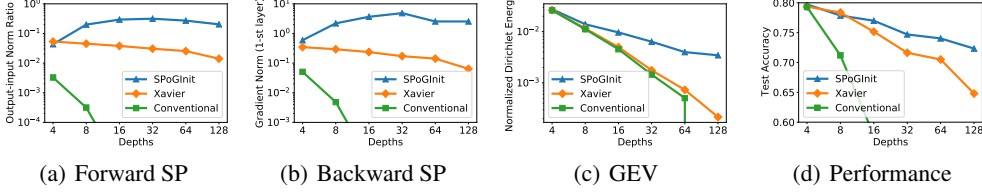

    (a) Forward SP         (b) Backward SP         (c) GEV         (d) Performance

Figure 4: (a) The forward metrics, (b) backward metrics, and (c) graph embedding variation metrics of deep GCNs with different initializations on the Cora dataset. (d) Test accuracies of deep GCNs after training on Cora. We find that SPoGInit simultaneously addresses three signal propagation aspects, and alleviates the performance degradation.

In Figure 4(a)-4(c), we report the average signal propagation metrics for vanilla GCNs with different initializations and varying depths. The results indicate that SPoGInit stabilizes the forward-backward propagations and enhances the graph embedding variation. Notably, SPoGInit successfully prevents gradient vanishing, a common issue encountered by other initialization. As a result, SPoGInit effectively alleviates the performance degradation of deep vanilla GCNs. It outperforms the baselines (Xavier, Conventional) by 7.5% and 35.2% test accuracy at depth 128 (see Figure 4(d)). Similar phenomena are also observed in various other datasets. We present more experiments in Appendix

---

[3]The design of gating parameters for residual networks coincides with Bachlechner et al. (2021).

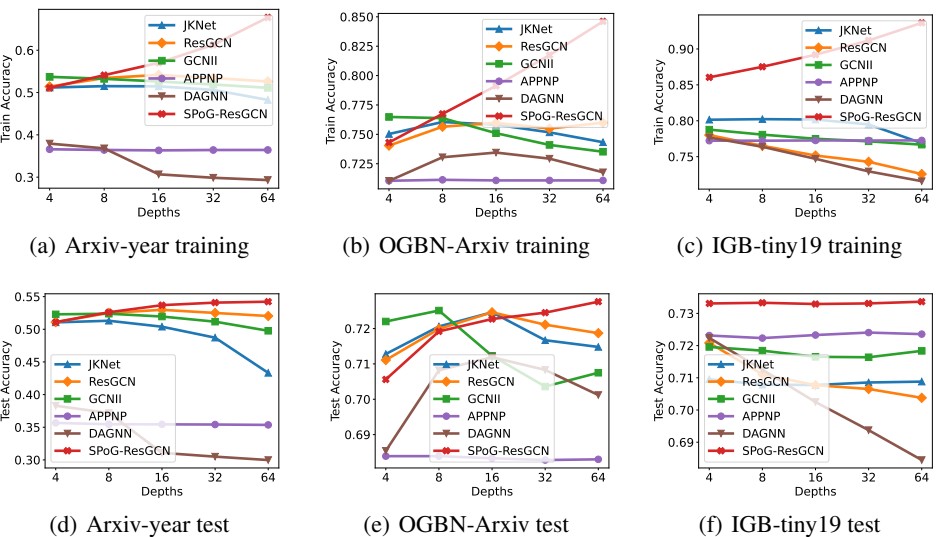

(a) Arxiv-year training     (b) OGBN-Arxiv training     (c) IGB-tiny19 training

(d) Arxiv-year test     (e) OGBN-Arxiv test     (f) IGB-tiny19 test

Figure 5: The average training accuracies (a)-(c) and test accuracies (d)-(e) of different skip-connection-based GCNs with ReLU activation on various datasets. SPoG-ResGCN outperforms baselines on all datasets and achieves consistent training gains with increasing depth
.

H.3. These results also demonstrate a strong correlation between the proposed signal propagation metrics and the actual performance of deep GCNs. To further verify the benefits of deep GCNs, We also provide experiments on the datasets with long-range dependencies in Appendix H.4.

## 5.2 EXPERIMENTS FOR SKIP-CONNECTION-BASED GCN MODELS

In this part, we consider GCN with skip-connections and examine **whether deep SPoG-ResGCN overcomes the curse of depth**. We adopt a few popular skip-connection-based GCN models, JKNet, ResGCN (with Conventional initialization), GCNII, APPNP, and DAGNN as baselines. Note that for small-sized datasets these baseline models performs very well already, thus we only consider large-scale datasets.

Figure 5 present the test and training accuracy of ReLU-activated models with various depths (experiments of tanh-activated models are presented in Appendix H.5). We see that SPoG-ResGCN achieves *consistent* performance gains as the depth increases. For instance, on the OGBN-Arxiv and Arxiv-year datasets, SPoG-ResGCN achieves a 2.2% gain in test accuracy as the model depth increases from 4 to 64. In contrast, for other models with standard initialization, the performance deteriorates as the depth increases.

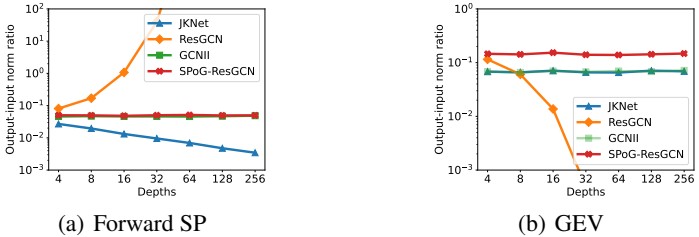

(a) Forward SP     (b) GEV

Figure 6: (a) The forward metrics and (b) graph embedding variation metrics of different models and depths on Cora. ResGCN suffers from forward exploding and GEV vanishing. In contrast, SPoG-ResGCN addresses the forward propagation and preserves the graph embedding variation.

Next, we investigate **whether SPoG-ResGCN achieves well-behaved signal propagation**. We adopt JKNet, ResGCN (with Conventional initialization), and GCNII as baselines. Figure 6 presents the average forward metric and graph embedding variation metric of different models with various

depths. Results indicate that SPoG-ResGCN effectively controls forward propagation and graph embedding variation.

Skip connections significantly change the back-propagation computation. Therefore, we select the middle hidden layer (the $L/2$-th layer in an $L$-layer model) as the representative layer to measure the backward propagation. Figure 7 plots the average backward metrics of the skip-connection-based GCNs with various depths $L$ during early training. We see that the baseline models suffer from poor backward propagation. Similar phenomena are also observed at initialization; see Appendix H.6. In contrast, the backward propagation of deep SPoG-ResGCN with different depths rapidly tends to stabilize during early training. The results of other layers in the skip-connection-based models are presented in Appendix H.6.

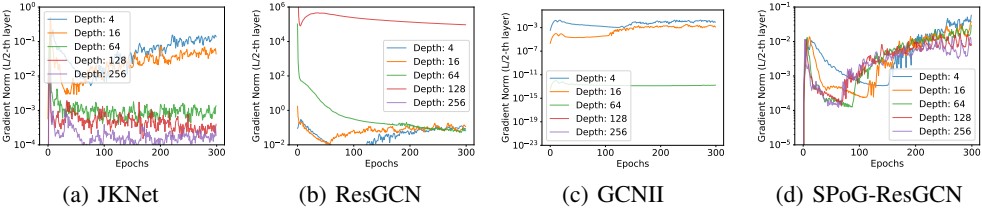

|    (a) JKNet    |    (b) ResGCN    |    (c) GCNII    |    (d) SPoG-ResGCN    |

Figure 7: The backward metrics at the $L/2$-th layer of different skip-connection-based GCNs with various depths $L$ during early training on the IGB-Tiny19 dataset. Baselines suffer from gradient vanishing or exploding problems. Across early 300 epochs, the gradient norms of (a) JKNet and (c) GCNII vanish as the depths increase, while the gradient norms of (b) ResGCN explode. In contrast, the gradient norms of (d) SPoG-ResGCN quickly improve in the early training. The disappearing lines in (b)-(c) are caused by surpassing the machine precision.

We summarize behaviors on signal propagation for skip-connection-based GCN models in Table 1.

Table 1: Summary of signal propagation for popular skip-connection-based GCNs. ✓ means that the corresponding signal propagation is well-behaved. The proposed SPoG-ResGCN addresses all three signal propagation aspects properly.

| Models | Forward SP | Backward SP | Graph embedding variation |
|---|---|---|---|
| JKNet | vanish | vanish | ✓ |
| ResGCN | explode | explode | vanish |
| GCNII | ✓ | vanish | ✓ |
| SPoG-ResGCN | ✓ | ✓ | ✓ |

In conclusion, SPoGInit offers a straightforward yet effective solution for ResGCNs to address the signal propagation challenges. As a result, deep SPoG-ResGCN possesses powerful training capability, successfully countering the curse of depth.

## 6 CONCLUSION

We attempt to address the performance degradatation of training deep GCNs from the lens of signal propagation. We consider three metrics: the forward propagation, backward propagation, and graph embedding variation propagation. Our theoretical analysis and empirical studies revealed that widely used initialization methods in GCNs fail to control these metrics simultaneously, resulting in performance degradation as depth increases. To tackle these challenges, a new initialization design method, termed SPoGInit, is proposed. Experiments demonstrate that SPoGInit effectively alleviates performance degradation in deep vanilla GCNs and deep ResGCNs. One interesting direction for future work is to study signal propagation and design initialization for GNNs with attention.

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

# Appendix

## Table of Contents

# A    RELATED WORKS

**Over-smoothing in GCNs.**    The over-smoothing issue was proposed in Li et al. (2018) to explain the curse of depth in deep GCNs and then studied in (Oono & Suzuki, 2019; Cai & Wang, 2020; Yang et al., 2020; Chen et al., 2020a; Rusch et al., 2023b; Luan et al., 2020; Cong et al., 2021; Zhang et al., 2022). Although the smoothing effects of graph convolution may benefit shallow GCNs (Keriven, 2022; Wu et al., 2023), they adversely affect the performance of deep GCNs. To alleviate over-smoothing, a variety of techniques are adopted (Chen et al., 2022b). For vanilla GCNs, techniques such as nodes or edges dropping (Srivastava et al., 2014; Zou et al., 2019; Rong et al., 2020; Huang et al., 2020; Lu et al., 2021), normalization (Ioffe & Szegedy, 2015; Zhao & Akoglu, 2020; Zhou et al., 2020; Yang et al., 2020; Zhou et al., 2021b; Li et al., 2020; Guo et al., 2023), and regularization (Chen et al., 2020a; Yang et al., 2020; Zhou et al., 2021a) were explored. Efforts were also taken on different variants of GCN architectures, including GCNs with residual connections (Kipf & Welling, 2017; Jaiswal et al., 2022), GCNs with jumping connections (Xu et al., 2018; Liu et al., 2020; Zhu et al., 2020), and so on (Bose & Das, 2023; Di Giovanni et al., 2022; Chien et al., 2021; Gasteiger et al., 2019; Luan et al., 2019; Chen et al., 2020b; Li et al., 2019; Yan et al., 2022; Guo et al., 2022; Min et al., 2020; Chen et al., 2022a; Jin et al., 2022; Zheng et al., 2021; Yang et al., 2023b; Li et al., 2021). In contrast to these existing works, our paper delves into the impact of weight initialization to tackle over-smoothing (as well as gradient pathology) in GCNs.

**Signal propagation.**    Classical signal propagation theory has built up a foundation for understanding how information flows through deep neural networks (DNNs) and guides the random weight initialization. At first, (Glorot & Bengio, 2010; He et al., 2015) studied the forward-backward propagation in linear or ReLU-activated models. Then, the mean-field theory (Neal, 1996; Lee et al., 2018; Matthews et al., 2018) was incorporated to study the signal propagation in models with general non-linear activation. Theoretical analysis on fully-connected neural networks (FCNNs) includes the study of Edge-of-Chaos (EOCs) (Poole et al., 2016; Schoenholz et al., 2017; Hayou et al., 2019; 2022) and dynamical isometry (Saxe et al., 2014; Pennington et al., 2017; 2018). Other works studied the signal propagation in deep CNN (Xiao et al., 2018), RNN (Chen et al., 2018), ResNet (Yang & Schoenholz, 2017; Hayou et al., 2022), autoencoder (Li & Nguyen, 2019), and LSTM/GRU (Gilboa et al., 2019). In the realm of GCNs, (Guo et al., 2022; Jaiswal et al., 2022) designed weight initialization methods via traditional forward and backward propagation. Our work further analyzes the graph embedding variation propagation. graph embedding variation propagation is specifically tailored for GCN-like architectures, and is shown to be crucial to resolving the curse of depth. In the realm of GCNs, (Guo et al., 2022; Jaiswal et al., 2022; Li et al., 2023) designed weight initialization methods based on forward and backward propagation. In contrast to these existing works, our paper additionally analyze the impact of initialization on graph embedding variation (GEV) to alleviate over-smoothing problem in GCNs, which enhance the power of deep GCNs and the performance on large-scale datasets.

**Weight searching and gating parameters.**    In addition to signal propagation, other factors that reflect the training dynamics have also been exploited to guide the searching of initial weights Dauphin & Schoenholz (2019); Zhu et al. (2021). Our SPoGInit draws inspiration from MetaInit (Dauphin & Schoenholz, 2019) and is further tailored to vanilla GCNs. For DNNs with residual connections, (De & Smith, 2020; Zhang et al., 2019; Bachlechner et al., 2021) introduced trainable gating parameters to preserve signal propagation. We borrow the idea from ReZero (Bachlechner et al., 2021) and propose SPoG-ResGCN, which incorporates skip connections and gating parameters in GCNs.

**Other works.**    Some existing works studied graph neural tangent kernel (GNTK) (Bayer et al., 2022; Du et al., 2019; Huang et al., 2022; Jiang et al., 2022; Sabanayagam et al., 2021; 2022; Zhou & Wang, 2022; Gebhart, 2022; Krishnagopal & Ruiz, 2023; Yang et al., 2023a). They analyzed the training dynamics of GCNs under the infinite-width limit.

## B  SUPPLEMENTAL NOTATION AND PRELIMINARIES

### B.1  NOTATION

For any integer $n \in \mathbb{N}$, we define $[n] \triangleq \{1, 2, \ldots, n\}$. We may denote a matrix $X \in \mathbb{R}^{m \times n}$ by $(x_{ij})_{i \in [m], j \in [n]}$, where $x_{ij}$ is the entry in the $i$-the row and the $j$-th column. We further use $X_{i,:} \in \mathbb{R}^{1 \times n}$ and $X_{:,j} \in \mathbb{R}^{m \times 1}$ to denote the $i$-th row and the $j$-th column of $X$, respectively. $\| \cdot \|_{\mathrm{F}}$ denotes the Frobenius norm. Given any function $f : \mathbb{R}^{m \times n} \to \mathbb{R}$, its derivative $\partial f / \partial X$ with respect to $X \in \mathbb{R}^{m \times n}$ is the $m \times n$ matrix with $(\partial f / \partial X)_{ij} = \partial f(X) / \partial x_{ij}$. For any activation function $\sigma : \mathbb{R} \to \mathbb{R}$, we use $\sigma(X) \in \mathbb{R}^{m \times n}$ to denote the output of applying $\sigma$ entry-wise to the matrix $X$, i.e., $(\sigma(X))_{ij} = \sigma(x_{ij})$. We denote ReLU activation by $\mathrm{ReLU}(x) \triangleq \max(0, x)$ and tanh activation by $\tanh(x) \triangleq (e^x - e^{-x})/(e^x + e^{-x})$. For brevity, we use $\theta$ to denote the collection of all trainable parameters in a GCN model.

For any matrix $X = (x_{ij}) \in \mathbb{R}^{m \times n}$, the vectorizaion of $X$ is defined by

$$\mathrm{vec}(X) := (x_{11}, \ldots, x_{m1}, x_{12}, \ldots, x_{m2}, \ldots, x_{1n}, \ldots, x_{mn})^T \in \mathbb{R}^{mn \times 1}.$$

For any matrix $X = (x_{ij}) \in \mathbb{R}^{m \times n}$ and $Y = (y_{ij}) \in \mathbb{R}^{p \times q}$, the Kronecker product of $X$ and $Y$ is a $mp \times nq$ block matrix defined by

$$X \otimes Y := \begin{pmatrix} x_{11}Y & \ldots & x_{1n}Y \\ \vdots & \ddots & \vdots \\ x_{m1}Y & \ldots & x_{mn}Y \end{pmatrix}.$$

For a matrix $X = (x_{ij}) \in \mathbb{R}^{m \times n}$, if $x_{ij} = 0$ for all $i \in [m]$ and $j \in [n]$, we denote $X = \mathbf{0}_{m \times n}$; if $x_{ij} = 1$ for all $i \in [m]$ and $j \in [n]$, we denote $X = \mathbf{1}_{m \times n}$. For a vector $Z = (z_i) \in \mathbb{R}^n$, if $z_i = 0$ for all $i \in [n]$, we denote $Z = \mathbf{0}_n$; if $z_i = 1$ for all $i \in [n]$, we denote $Z = \mathbf{1}_n$.

### B.2  SUPPLEMENTAL SKIP-CONNECTED-BASED GCN ARCHITECTURES

**JKNet.**  Xu et al. (2018) proposes jumping knowledge network (JKNet) by only combining all embeddings in the hidden layers before getting the output. To be more specific, an $L$-layer JKNet is defined by

$$X^{(0)} := XW^{(0)} + \mathbf{1}_n \cdot b^{(0)},$$
$$H^{(l)} := \hat{A}X^{(l-1)}W^{(l)} + \mathbf{1}_n \cdot b^{(l)}, \quad \text{for any } l \in [L],$$
$$X^{(l)} := \sigma(H^{(l)}), \quad \text{for any } l \in [L],$$
$$H^{(\mathrm{out}, L)} := \mathrm{COMBINE}(X^{(1)}, X^{(2)}, \ldots, X^{(L)}).$$

We assume that COMBINE is a linear transformation from the concatenation of $\{X^{(l)}\}_{l=1}^L$ to the output embedding.

**GCNII.**  Chen et al. (2020b) designs GCNII by (1) using residual connection to the initial layer and (2) combining identity matrices with the weight matrices. Specifically, an $L$-layer GCNII is defined by

$$X^{(0)} := XW^{(0)} + \mathbf{1}_n \cdot b^{(0)},$$
$$H^{(l)} := (1 - \alpha_l)\hat{A}X^{(l-1)} \cdot \left[(1 - \beta_l)I_d + \beta_l W_1^{(l)}\right]$$
$$+ \alpha_l X^{(0)} \cdot \left[(1 - \beta_l)I_d + \beta_l W_2^{(l)}\right], \quad \text{for any } l \in [L], \quad (3)$$
$$X^{(l)} := \sigma(H^{(l)}), \quad \text{for any } l \in [L],$$
$$H^{(\mathrm{out}, L)} := X^{(L)}W^{(L+1)} + \mathbf{1}_n \cdot b^{(L+1)},$$

where $\{\alpha_l, \beta_l\}_{l=1}^L$ are predetermined hyperparameters and $\beta_l$, set to be $\log(\frac{\lambda}{l+1} + 1)$, vanishes to 0 as $l \to \infty$, while $\lambda$ is a predetermined hyperparameter. Chen et al. (2020b) call the architecture imposing $W_1^{(l)} = W_2^{(l)}$ by GCNII and call its improved version by (3) GCNII* in their paper. For the sake of brevity, we refer to the architecture (3) as GCNII without bringing any confusion.

## C  CONVOLUTIONAL KERNEL

Suppose that graph $\mathcal{G}$ has $M$ connected components. The $m$-th component is a subgraph denoted by $\mathcal{G}_m = (\mathcal{V}_m, \mathcal{E}_m)$ for $m \in [M]$. We present a well-known result characterizing the eigenvalues and the eigenvectors of $\hat{A}$ without giving proof, see, e.g., Proposition 1 in Oono & Suzuki (2019).

**Proposition C.1.** *Suppose that $\mathcal{G} = (\mathcal{V}, \mathcal{E})$ has $M$ connected components $\{\mathcal{G}_m = (\mathcal{V}_m, \mathcal{E}_m)\}_{m=1}^M$ and the eigenvalues of $\hat{A}$ are $\lambda_1 \geq \lambda_2 \geq \cdots \geq \lambda_n$. Then we have*

- $\lambda_i = 1$, *for any* $1 \leq i \leq M$.

- $\lambda_i \in (-1, 1)$, *for any* $M + 1 \leq i \leq n$.

*Moreover, the set $\{v^{(m)} = \tilde{D}^{\frac{1}{2}} u^{(m)} : m \in [M]\}$ is a basis of the $m$-dimensional eigenspace of $\hat{A}$ corresponding to the eigenvalue 1, where $u^{(m)} = (\mathbb{1}_{\{i \in \mathcal{V}_m\}})_{i \in [n]} \in \mathbb{R}^{n \times 1}$ is the indicator vector of the $m$-th connected component $\mathcal{G}_m$.*

**Lemma C.2.** *Given any $H \in \mathbb{R}^{n \times C}$ and $H \neq \mathbf{0}_{n \times C}$, we have $0 \leq \mathrm{Dir}(H)/\|H\|_{\mathrm{F}}^2 \leq 2$.*

*Proof.* Recall that $\hat{L} = I - \hat{A}$ is the normalized Laplacian of graph $\mathcal{G}$. By Proposition C.1, all the eigenvalues of $\hat{L}$ belong to $[0, 2)$.

Given any $H \in \mathbb{R}^{n \times C}$, we have

$$\mathrm{Dir}(H) = \mathrm{tr}(H^T \hat{L} H) = \sum_{k=1}^C H_{:,k}^T \hat{L} H_{:,k} \leq \sum_{k=1}^C 2 \cdot H_{:,k}^T H_{:,k} = 2\|H\|_{\mathrm{F}}^2.$$

Similarly, we have

$$\mathrm{Dir}(H) = \mathrm{tr}(H^T \hat{L} H) = \sum_{k=1}^C H_{:,k}^T \hat{L} H_{:,k} \geq \sum_{k=1}^C 0 \cdot H_{:,k}^T H_{:,k} = 0.$$

Therefore, we conclude that

$$0 \leq \mathrm{Dir}(H)/\|H\|_{\mathrm{F}}^2 \leq 2.$$

$\square$

# D  SIGNAL PROPAGATION THEORY FOR VANILLA GCN

## D.1  NNGP CORRESPONDENCE FOR VANILLA GCN

**Proposition D.1** (NNGP correspondence for vanilla GCN). *As the network widths $d_1, d_2, \ldots, d_{L-1}$ sequentially go to infinity, the $l$-th layer's pre-activation embedding channels $\{H^{(l)}_{:,k}\}_{k \in [d_l]}$ converge to i.i.d. $n$-dimensional Gaussian random variables $N(\mathbf{0}_n, \Sigma^{(l)})$ in distribution for any $l \geq 2$. The covariance matrices are*

$$\Sigma^{(1)} = \frac{\sigma_w^2}{d_0} \hat{A} X X^T \hat{A},$$
$$\Sigma^{(l+1)} = \sigma_w^2 \hat{A} G(\Sigma^{(l)}) \hat{A},$$

(4)

*where $G(\Sigma) = \mathbb{E}_{h \sim N(\mathbf{0}_n, \Sigma)}[\sigma(h)\sigma(h)^T]$ for any $n \times n$ positive semi-definite matrix $\Sigma$.*

*Proof of Proposition D.1.* We will prove that $\{H^{(l)}_{:,k}\}_{k \in [d_l]}$ are asymptotically i.i.d. $n$-dimensional random variables with mean $\mathbf{0}_n$ and covariance matrix $\Sigma^{(l)}$ for any $l \geq 1$ under the infinite width limit by mathematical induction. Proposition D.1, which contains a stronger claim that $\{H^{(l)}_{:,k}\}$ are asymptotically Gaussian for any $l \geq 2$, will be shown during the induction steps.

**Base case.** Since the bias terms are initialized to be zero, when $l = 1$, the $k$-th channel of the embedding is

$$H^{(1)}_{:,k} = \hat{A} X W^{(1)}_{:,k} + \mathbf{1}_n \cdot b^{(1)}_k = \hat{A} X W^{(1)}_{:,k}.$$

(5)

Since $\{W^{(1)}_{:,k}\}_{k \in [d_1]}$ are i.i.d. random variables, so $\{H^{(1)}_{:,k}\}_{k \in [d_1]}$ are also i.i.d. random variables. Taking the expectation of (5), we get

$$\mathbb{E}[H^{(1)}_{:,k}] = \hat{A} X \cdot \mathbb{E}[W^{(1)}_{:,k}] = \mathbf{0}_n.$$

Calculating the covariance matrix of (5), we have

$$\begin{aligned}
\mathrm{Cov}[H^{(1)}_{:,k}, H^{(1)}_{:,k}] &= \mathbb{E}[H^{(1)}_{:,k} \cdot H^{(1)T}_{:,k}] = \mathbb{E}[\hat{A} X W^{(1)}_{:,k} W^{(1)T}_{:,k} X^T \hat{A}] \\
&= \hat{A} X \cdot \mathbb{E}[W^{(1)}_{:,k} W^{(1)T}_{:,k}] \cdot X^T \hat{A} = \hat{A} X \cdot \left( \frac{\sigma_w^2}{d_0} \cdot I_{d_0} \right) \cdot X^T \hat{A} \\
&= \frac{\sigma_w^2}{d_0} \hat{A} X X^T \hat{A}.
\end{aligned}$$

Thus, if we define $\Sigma^{(1)} = \sigma_w^2 \hat{A} X X^T \hat{A}/d_0$, then $\{H^{(1)}_{:,k}\}_{k \in [d_1]}$ are exactly i.i.d with mean $\mathbf{0}_n$ and covariance matrix $\Sigma^{(1)}$.

**Induction step.** Suppose that $\{H^{(l)}_{:,k}\}_{k \in [d_l]}$ converge to i.i.d. $n$-dimensional random variables with mean $\mathbf{0}_n$ and covariance matrix $\Sigma^{(l)}$ in distribution as $d_1, \ldots, d_{l-1}$ sequentially go to infinity, we look at the $(l+1)$-th layer. Recall from the formation of the $l$-th layer in vanilla GCN, we have

$$\begin{aligned}
H^{(l+1)} &= \hat{A} X^{(l)} W^{(l+1)} + \mathbf{1}_n \cdot b^{(l+1)}, \\
X^{(l)} &= \sigma(H^{(l)}),
\end{aligned}$$

for any $l \geq 1$. We vectorize the first equation and get

$$\begin{aligned}
\mathrm{vec}(H^{(l+1)}) &= \mathrm{vec}(\hat{A} X^{(l)} W^{(l+1)}) + \mathrm{vec}(\mathbf{1}_n \cdot b^{(l+1)}) \\
&= \sum_{k=1}^{d_l} \mathrm{vec}\left( \underbrace{[\hat{A} X^{(l)}_{:,k}]}_{n \times 1} \cdot \underbrace{W^{(l+1)}_{k,:}}_{1 \times d_{l+1}} \right),
\end{aligned}$$

(6)

because $b^{(l+1)}$ is initialized to be $\mathbf{0}_{d_{l+1}}$. Suppose that $\Sigma^{(l+1)} = \sigma_w^2 \hat{A} G(\Sigma^{(l)}) \hat{A}$, we are going to show that $\mathrm{vec}(H^{(l+1)})$ converges to a Gaussian random variable $N(\mathbf{0}_{nd_{l+1}}, I_{d_{l+1}} \otimes \Sigma^{(l+1)})$ in distribution as $d_1, d_2, \ldots, d_{l-1}, d_l$ sequentially go to infinity. If this claim holds, $\{H^{(l+1)}_{:,k}\}$ are

not only asymptotically i.i.d., but also asymptotically Gaussian i.i.d. with $N(\mathbf{0}_n, \Sigma^{(l+1)})$, which corresponds to the statement of this proposition.

For brevity, we define

$$\omega_{kk'}^{(l+1)} := \sqrt{d_l} \cdot W_{kk'}^{(l+1)}, \quad \text{for all } k \in [d_l] \text{ and } k' \in [d_{l+1}],$$

and

$$Z_k^{(l+1)} := \text{vec}\left([\hat{A}X_{:,k}^{(l)} \cdot \omega_{k,:}^{(l+1)}]\right), \quad \text{for all } k \in [d_l]. \tag{7}$$

Then we get that $\{\omega_{kk'}^{(l+1)}\}_{k \in [d_l], k' \in [d_{l+1}]}$ are i.i.d. from $N(0, \sigma_w^2)$ and

$$\text{RHS of (6)} = \frac{1}{\sqrt{d_l}} \sum_{k=1}^{d_l} Z_k^{(l+1)}. \tag{8}$$

By the induction hypothesis, as $d_1, d_2 \ldots, d_{l-1}$ sequentially go to infinity, $\{X_{:,k}^{(l)}\}_{k \in [d_l]} = \{\sigma(H_{:,k}^{(l)})\}_{k \in [d_l]}$ converge to i.i.d. $n$-dimensional random vectors in distribution. Because $X^{(l)}$ can be regarded as a function of $\{W^{(l')}\}_{l'=1}^{l}$ at initialization, we get that $X^{(l)}$ and $W^{(l+1)}$ are independent. Thus, as $d_1, d_2 \ldots, d_{l-1}$ sequentially go to infinity, $\{Z_k^{(l+1)}\}_{k \in [d_l]}$ converge to i.i.d. random vectors in distribution. Moreover, in this limiting case, by taking the expectation of (7), we have

$$\mathbb{E}[Z_1^{(l+1)}] = \text{vec}\left(\left[\hat{A}\mathbb{E}[X_{:,k}^{(l)}]\right] \cdot \mathbb{E}[\omega_{k,:}^{(l+1)}]\right) = \text{vec}\left(\mathbf{0}_{n\times 1} \cdot \mathbf{0}_{1\times d_{l+1}}\right) = \mathbf{0}_{nd_{l+1}}.$$

Calculating the covariance matrix of (7), we have

$$\begin{aligned}
\text{Cov}[Z_1^{(l+1)}, Z_1^{(l+1)}] &= \mathbb{E}[Z_1^{(l+1)} \cdot Z_1^{(l+1)T}] \\
&= \mathbb{E}\left[\text{vec}\left([\hat{A}X_{:,1}^{(l)} \cdot \omega_{1,:}^{(l+1)}]\right) \cdot \text{vec}\left([\hat{A}X_{:,1}^{(l)} \cdot \omega_{1,:}^{(l+1)}]\right)^T\right] \\
&= \mathbb{E}\left[(\omega_{1,:}^{(l+1)T} \otimes \hat{A}X_{:,1}^{(l)}) \cdot (\omega_{1,:}^{(l+1)} \otimes X_{:,1}^{(l)T}\hat{A})\right] \\
&= \mathbb{E}\left[\omega_{1,:}^{(l+1)T} \omega_{1,:}^{(l+1)} \otimes \hat{A}X_{:,1}^{(l)}X_{:,1}^{(l)T}\hat{A}\right] \\
&= \mathbb{E}\left[\omega_{1,:}^{(l+1)T} \omega_{1,:}^{(l+1)}\right] \otimes \left\{\hat{A} \cdot \mathbb{E}\left[X_{:,1}^{(l)}X_{:,1}^{(l)T}\right] \cdot \hat{A}\right\} \\
&= \sigma_w^2 I_{d_{l+1}} \otimes \hat{A}G(\Sigma^{(l)})\hat{A} \\
&= I_{d_{l+1}} \otimes \sigma_w^2 \hat{A}G(\Sigma^{(l)})\hat{A} = I_{d_{l+1}} \otimes \Sigma^{(l+1)}.
\end{aligned}$$

Here $X_{:,1}^{(l)}$ actually stands for the limit of true $X_{:,1}^{(l)}$ as $d_1, \ldots, d_{l-1}$ sequentially go to infinity without bringing any confusion.

By multivariate central limit theorem, $\frac{1}{\sqrt{d_l}} \sum_{k=1}^{d_l} Z_k^{(l+1)}$ converges to a Gaussian random variable $N(\mathbf{0}_{nd_{l+1}}, I_{d_{l+1}} \otimes \Sigma^{(l+1)})$ in distribution as $d_l \to \infty$. Recalling (6) and (8), we conclude that $\text{vec}(H^{(l+1)})$ converges to a Gaussian random variable $N(\mathbf{0}_{nd_{l+1}}, I_{d_{l+1}} \otimes \Sigma^{(l+1)})$ as $d_1, \ldots, d_l$ sequentially go to infinity.

**Conclusion.** By the principle of mathematical induction, we have proven this proposition. $\square$

## D.2 SOME DISCUSSION W.R.T. $G$

We claim that the function $G$ is well-defined in Proposition D.1 on the collection of positive semi-definite matrices

$$\mathcal{S} = \{\Sigma \in \mathbb{R}^{n\times n} : x^T \Sigma x \geq 0 \text{ for all } x \in \mathbb{R}^{n\times 1}\}. \tag{9}$$

*Remark* D.2. To show that $G(\Sigma) = \mathbb{E}_{h\sim N(\mathbf{0}_n, \Sigma)}[\sigma(h)\sigma(h)^T]$ is well-defined at any $\Sigma \in \mathcal{S}$, we only need to show that such $\Sigma$ is always a feasible covariance matrix of Gaussian distribution. For any $\Sigma \in \mathcal{S}$, there exists $P \in \mathbb{R}^{n\times n}$, such that $PP^T = \Sigma$. Let $\xi \sim N(\mathbf{0}_n, I_n)$ be an $n$-dimensional standard normal random variable, then the random variable $P\xi \sim N(\mathbf{0}_n, \Sigma)$. Thus, all positive semi-definite matrices are feasible covariance matrices for Gaussian distributions.

**Definition D.3.** Given any positive semi-definite matrix $\Sigma \in \mathcal{S}$, we define

$$G_1(\Sigma) := q(\Sigma)q(\Sigma)^T, \tag{10}$$

where $q(\Sigma) \in \mathbb{R}^{n \times 1}$ is defined by

$$q(\Sigma)_i := \sqrt{G(\Sigma)_{ii}}, \quad \text{for all } i \in [n]. \tag{11}$$

**Lemma D.4.** *Given any positive semi-definite matrix $\Sigma \in \mathcal{S}$, it holds that*

$$G_1(\Sigma)_{ij} \geq G(\Sigma)_{ij} \quad \text{for any } i, j \in [n]. \tag{12}$$

*Proof.* Recalling the formation of function $G$ in Proposition D.1 (NNGP correspondence for vanilla GCN), for any $i, j \in [n]$, we have

$$G(\Sigma)_{ij} = \mathbb{E}_{h \sim N(\mathbf{0}_n, \Sigma)}[\sigma(h_i) \cdot \sigma(h_j)].$$

Recalling (10) and (11) in Definition D.3, we get

$$\begin{aligned} G_1(\Sigma)_{ij} &:= q(\Sigma)_i \cdot q(\Sigma)_j = \sqrt{G(\Sigma)_{ii}} \cdot \sqrt{G(\Sigma)_{jj}} \\ &= \mathbb{E}_{h \sim N(\mathbf{0}_n, \Sigma)}[\sigma(h_i)^2]^{\frac{1}{2}} \cdot \mathbb{E}_{h \sim N(\mathbf{0}_n, \Sigma)}[\sigma(h_j)^2]^{\frac{1}{2}} \end{aligned} \tag{13}$$

From Hölder's inequality (Hardy et al., 1952), we get

$$\begin{aligned} \text{RHS of } (13) &\geq \mathbb{E}_{h \sim N(\mathbf{0}_n, \Sigma)}\left[|\sigma(h_i) \cdot \sigma(h_j)|\right] \\ &\geq \mathbb{E}_{h \sim N(\mathbf{0}_n, \Sigma)}\left[\sigma(h_i) \cdot \sigma(h_j)\right] = G(\Sigma)_{ij}. \end{aligned}$$

$\square$

**Lemma D.5.** *Given the NNGP covariance matrices $\{\Sigma^{(l)}\}_{l=1}^{\infty}$ defined by (4), it holds that*

$$\text{tr}(\Sigma^{(l+1)}) \leq \sigma_w^2 \, \text{tr}(G(\Sigma^{(l)})).$$

*Proof.* Recalling the NNGP correspondence formula for vanilla GCN (4) in Proposition D.1, we have

$$\text{tr}(\Sigma^{(l+1)}) = \text{tr}(\sigma_w^2(\hat{A}G(\Sigma^{(l)})\hat{A})) = \sigma_w^2 \, \text{tr}(\hat{A}G(\Sigma^{(l)})\hat{A}). \tag{14}$$

Since all entries of $\hat{A}$ are non-negative, by Lemma D.4, we have

$$(\hat{A}G(\Sigma^{(l)})\hat{A})_{ii} \leq (\hat{A}G_1(\Sigma^{(l)})\hat{A})_{ii}, \quad \text{for any } i \in [n].$$

Taking the summation of w.r.t $i \in [n]$, we get

$$\text{tr}(\hat{A}G(\Sigma^{(l)})\hat{A}) \leq \text{tr}(\hat{A}G_1(\Sigma^{(l)})\hat{A}). \tag{15}$$

Recalling the definition of function $G_1$ in (10), we get

$$\text{tr}(\hat{A}G_1(\Sigma^{(l)})\hat{A}) = \text{tr}(\hat{A}q(\Sigma^{(l)})q(\Sigma^{(l)})^T\hat{A}) = \|\hat{A}q(\Sigma^{(l)})\|^2. \tag{16}$$

By Proposition C.1, all the eigenvalues of $\hat{A}$ belong to $(-1, 1]$. Recalling the definition of function $q$ in (11), we get

$$\|\hat{A}q(\Sigma^{(l)})\|^2 \leq \|q(\Sigma^{(l)})\|^2 = \sum_{i=1}^{n} q(\Sigma^{(l)})_i^2 = \text{tr}(G(\Sigma^{(l)})). \tag{17}$$

Finally, combining (14), (15), (16), and (17), we complete the proof.

$\square$

### D.3 PROOF OF THEOREM 3.1 (SIGNAL PROPAGATION ON RELU-LIKE-ACTIVATED VANILLA GCN)

We will give a more general signal propagation analysis on vanilla GCN with ReLU-like activation.

**Definition D.6** (ReLU-like activation). An activation function $\sigma : \mathbb{R} \to \mathbb{R}$ is $(\alpha, \beta)$-ReLU if it has the form

$$\sigma(x) = \begin{cases} \alpha x, & x \geq 0, \\ \beta x, & x < 0, \end{cases} \tag{18}$$

where $\alpha, \beta \in \mathbb{R}_+$ and not both of them are 0. We also call such $\sigma$ a ReLU-like activation function.

Then we extend our analysis from the special $(1, 0)$-ReLU-activated case to the general $(\alpha, \beta)$-ReLU-activated case.

**Theorem D.7** (The generalized version of Theorem 3.1). *Under the NNGP correspondence approximation, when the activation function $\sigma$ is $(\alpha, \beta)$-ReLU in Definition D.6, we have*

*1. When $\sigma_w^2 = 2/(\alpha^2 + \beta^2)$, either the graph embedding variation metric*

$$\lim_{L \to \infty} \mathbf{M}_{GEV}^{(L)}(\sigma_w^2) = \lim_{L \to \infty} \mathbb{E}_{H \sim N(\mathbf{0}_n, \Sigma^{(L)})} \left[ \mathrm{Dir}(H)/\|H\|_{\mathrm{F}}^2 \right] = 0,$$

*or the forward propagation metric*

$$\lim_{L \to \infty} \mathbf{M}_{FSP}^{(L)}(\sigma_w^2) = \lim_{L \to \infty} \mathbb{E}_{H \sim N(\mathbf{0}_n, \Sigma^{(L)})} [\|H\|_{\mathrm{F}}^2/\|X\|_{\mathrm{F}}^2] = 0.$$

*2. When $\sigma_w^2 < 2$, for any $L \geq 1$, the forward propagation metric satisfies*

$$\mathbf{M}_{FSP}^{(L)}(\sigma_w^2) = \mathbb{E}_{H \sim N(\mathbf{0}_n, \Sigma^{(L)})} [\|H\|_{\mathrm{F}}^2/\|X\|_{\mathrm{F}}^2] \leq \frac{2C}{(\alpha^2 + \beta^2) d_0} \cdot \left( \frac{\sigma_w^2 (\alpha^2 + \beta^2)}{2} \right)^L.$$

**Lemma D.8.** *For any $x \in \mathbb{R}^n$, it holds that*

$$\mathrm{Dir}(\hat{A}x) \leq \lambda^2 \mathrm{Dir}(x), \tag{19}$$

*where $\lambda$ is the second largest absolute eigenvalue of $\hat{A}$, i.e.,*

$$\lambda = \max_{i \in [n], \lambda_i \neq 1} |\lambda_i|.$$

*Proof.* Since $\hat{A}$ is a symmetric real matrix, by Proposition C.1, it can be decomposed as $\hat{A} = U \Lambda U^T$, where $\Lambda = \mathrm{diag}(\lambda_1, \lambda_2, \ldots, \lambda_n)$ and $U \in \mathbb{R}^{n \times n}$ is an orthogonal matrix. The $i$-th column $u_i$ of $U$ is the eigenvector corresponding to $\lambda_i$.

By Proposition C.1, we have $\lambda_i \in (-1, 1]$ for all $i \in [n]$. Since $\hat{L} = I - \hat{A}$, we conclude that

$$\mathrm{Dir}(\hat{A}x) = (\hat{A}x)^T \hat{L} \hat{A}x = x^T \hat{A} \hat{L} \hat{A}x = z^T U^T (U \Lambda U^{-1})(U(I - \Lambda)U^{-1})(U \Lambda U^{-1})z$$

$$= z^T \Lambda (I - \Lambda) \Lambda z = \sum_{i=1}^n (1 - \lambda_i) \lambda_i^2 z_i^2 \leq \lambda^2 \sum_{i=1}^n (1 - \lambda_i) z_i^2$$

$$= \lambda^2 z^T (I - \Lambda) z = \lambda^2 \mathrm{Dir}(x).$$

$\square$

**Lemma D.9.** *When the activation function $\sigma$ is $(\alpha, \beta)$-ReLU, it holds that*

$$(\sigma(x) - \sigma(y))^2 + (\sigma(-x) - \sigma(-y))^2 \leq (\alpha^2 + \beta^2)(x - y)^2, \tag{20}$$

*for any $x, y \in \mathbb{R}$. Moreover, the inequality becomes an equality if and only if $xy \geq 0$.*

*Proof.* When $x, y \geq 0$, it holds that

$$\text{LHS of } (20) = (\alpha x - \alpha y)^2 + (-\beta x + \beta y)^2 = \text{RHS of } (20).$$

Similarly, the equality holds when $x, y \leq 0$. When $xy < 0$,

$$
\begin{aligned}
\text{LHS of (20)} &= (\alpha x - \beta y)^2 + (-\beta x + \alpha y)^2 \\
&= (\alpha^2 + \beta^2)(x^2 + y^2) - 4\alpha\beta xy \\
&= (\alpha^2 + \beta^2)(x - y)^2 + 2(\alpha - \beta)^2 xy \\
&< \text{RHS of (20)}.
\end{aligned}
$$

$\square$

**Lemma D.10.** *When the activation function $\sigma$ is $(\alpha, \beta)$-ReLU, it holds that*

$$
\mathrm{Dir}(\sigma(h)) + \mathrm{Dir}(\sigma(-h)) \leq (\alpha^2 + \beta^2)\mathrm{Dir}(h). \tag{21}
$$

*Proof.* Since the activation function $\sigma$ is $(\alpha, \beta)$-ReLU, we have

$$
\sigma(cx) = c\sigma(x), \quad \text{for any } c \in \mathbb{R}_+, x \in \mathbb{R}.
$$

Then we get

$$
\text{LHS of (21)} = \sum_{(i,j) \in \mathcal{E}} \left[ \frac{\sigma(h_i)}{\sqrt{1 + d_i}} - \frac{\sigma(h_j)}{\sqrt{1 + d_j}} \right]^2 + \left[ \frac{\sigma(-h_i)}{\sqrt{1 + d_i}} - \frac{\sigma(-h_j)}{\sqrt{1 + d_j}} \right]^2
$$

$$
= \sum_{(i,j) \in \mathcal{E}} \left[ \sigma\left( \frac{h_i}{\sqrt{1 + d_i}} \right) - \sigma\left( \frac{h_j}{\sqrt{1 + d_j}} \right) \right]^2 + \left[ \sigma\left( \frac{-h_i}{\sqrt{1 + d_i}} \right) - \sigma\left( \frac{-h_j}{\sqrt{1 + d_j}} \right) \right]^2.
$$

By Lemma D.9, we have

$$
\text{LHS of (21)} \leq (\alpha^2 + \beta^2) \sum_{(i,j) \in \mathcal{E}} \left[ \frac{h_i}{\sqrt{1 + d_i}} - \frac{h_j}{\sqrt{1 + d_j}} \right]^2 = \text{RHS of (21)}.
$$

$\square$

**Lemma D.11.** *When the activation function $\sigma$ is $(\alpha, \beta)$-ReLU, for any feasible covariance matrix $\Sigma \in \mathbb{R}^{n \times n}$, it holds that*

$$
\mathbb{E}_{h \sim N(\mathbf{0}_n, \Sigma)}[\mathrm{Dir}(\sigma(h))] \leq \frac{\alpha^2 + \beta^2}{2} \cdot \mathbb{E}_{h \sim N(\mathbf{0}_n, \Sigma)}[\mathrm{Dir}(h)].
$$

*Proof.* By symmetry, for any $n$-dimensional random variable $h \sim N(\mathbf{0}_n, \Sigma)$, it holds that $-h \sim N(\mathbf{0}_n, \Sigma)$. By Lemma D.10, we have

$$
\begin{aligned}
2\mathbb{E}_{h \sim N(\mathbf{0}_n, \Sigma)}[\mathrm{Dir}(\sigma(h))] &= \mathbb{E}_{h \sim N(\mathbf{0}_n, \Sigma)}[\mathrm{Dir}(\sigma(h)) + \mathrm{Dir}(\sigma(-h))] \\
&\leq (\alpha^2 + \beta^2)\mathbb{E}_{h \sim N(\mathbf{0}_n, \Sigma)}[\mathrm{Dir}(h)].
\end{aligned}
$$

$\square$

**Lemma D.12.** *Under the NNGP correspondence approximation, suppose that the activation function $\sigma$ is $(\alpha, \beta)$-ReLU in Definition D.6. If*

$$
\sigma_w^2 < \frac{2}{\lambda^2(\alpha^2 + \beta^2)},
$$

*then we have*

$$
\mathbb{E}_{h \sim N(\mathbf{0}_n, \Sigma^{(l)})}[\mathrm{Dir}(h)] = O\left( \left( \frac{\lambda^2 \sigma_w^2 (\alpha^2 + \beta^2)}{2} \right)^l \right), \quad \text{as } l \to \infty,
$$

*where $\lambda$ is the second largest non-one absolute eigenvalue of $\hat{A}$, i.e.,*

$$
\lambda = \max_{i \in [n], \lambda_i \neq 1} |\lambda_i|.
$$

*Proof.* For any positive semi-definite matrix $\Sigma \in \mathcal{S}$ and any $n$-dimensional Gaussian random variable $h \sim N(\mathbf{0}_n, \Sigma)$, we have

$$\mathbb{E}_{h \sim N(\mathbf{0}_n, \Sigma)}[\mathrm{Dir}(h)] = \mathbb{E}_{h \sim N(\mathbf{0}_n, \Sigma)}[\mathrm{tr}(h^T \hat{L} h)] = \mathbb{E}_{h \sim N(\mathbf{0}_n, \Sigma)}[\mathrm{tr}(\hat{L} h h^T)] = \mathrm{tr}(\hat{L}\Sigma).$$

Then according the NNGP correspondence formula (4) in Proposition D.1, for any $l \in \mathbb{N}$, we have

$$
\begin{aligned}
&\mathbb{E}_{h \sim N(\mathbf{0}_n, \Sigma^{(l+1)})}[\mathrm{Dir}(h)] = \mathrm{tr}(\hat{L}\Sigma^{(l+1)}) \\
&= \sigma_w^2 \, \mathrm{tr}(\hat{L}\hat{A}G(\Sigma^{(l)})\hat{A}) = \sigma_w^2 \, \mathrm{tr}\left(\hat{L}\hat{A} \cdot \mathbb{E}_{h \sim N(\mathbf{0}_n, \Sigma^{(l)})}[\sigma(h)\sigma(h)^T] \cdot \hat{A}\right) \\
&= \sigma_w^2 \mathbb{E}_{h \sim N(\mathbf{0}_n, \Sigma^{(l)})}\left[\mathrm{tr}\left(\hat{L}\hat{A}\sigma(h)\sigma(h)^T\hat{A}\right)\right] = \sigma_w^2 \mathbb{E}_{h \sim N(\mathbf{0}_n, \Sigma^{(l)})}\left[\mathrm{tr}\left(\sigma(h)^T\hat{A}\hat{L}\hat{A}\sigma(h)\right)\right] \\
&= \sigma_w^2 \mathbb{E}_{h \sim N(\mathbf{0}_n, \Sigma^{(l)})}\left[\mathrm{Dir}\left(\hat{A}\sigma(h)\right)\right].
\end{aligned}
\tag{22}
$$

By Lemma D.8 and Lemma D.11, we get

$$\text{RHS of (22)} \le \lambda^2 \sigma_w^2 \cdot \mathbb{E}_{h \sim N(\mathbf{0}_n, \Sigma^{(l)})}[\mathrm{Dir}(\sigma(h))] \le \frac{\lambda^2 \sigma_w^2 (\alpha^2 + \beta^2)}{2} \cdot \mathbb{E}_{h \sim N(\mathbf{0}_n, \Sigma^{(l)})}[\mathrm{Dir}(h)].
\tag{23}$$

Thus, combining (22) and (23), by induction, we have

$$\mathbb{E}_{h \sim N(\mathbf{0}_n, \Sigma^{(l)})}[\mathrm{Dir}(h)] = O\left(\left(\frac{\lambda^2 \sigma_w^2 (\alpha^2 + \beta^2)}{2}\right)^l\right), \quad \text{as } l \to \infty.$$

$\square$

*Proof of Theorem D.7 (the generalized version of Theorem 3.1).* First of all, we will prove **part 2** of this theorem. For any positive semi-definite matrix $\Sigma \in \mathcal{S}$, we have

$$\mathbb{E}_{h \sim N(\mathbf{0}_n, \Sigma)}[\|h\|^2] = \mathbb{E}_{h \sim N(\mathbf{0}_n, \Sigma)}[\mathrm{tr}(h^T h)] = \mathbb{E}_{h \sim N(\mathbf{0}_n, \Sigma)}[\mathrm{tr}(h h^T)] = \mathrm{tr}(\Sigma).$$

For this reason, we only need to focus on $\{\mathrm{tr}(\Sigma^{(l)})\}_{l=1}^{\infty}$ in the following proof.

We will show that $\{\mathrm{tr}(\Sigma^{(l)})\}_{l=1}^{\infty}$ is a decreasing sequence if $\sigma_w \le 2/(\alpha^2 + \beta^2)$. By Lemma D.5, we have

$$\mathrm{tr}(\Sigma^{(l+1)}) \le \sigma_w^2 \, \mathrm{tr}(G(\Sigma^{(l)})).
\tag{24}$$

When the activation function $\sigma$ is $(\alpha, \beta)$-ReLU, for any $c \in \mathbb{R}_+$, it holds that

$$
\begin{aligned}
\mathbb{E}_{Z \sim N(0,1)}[\sigma(cZ)^2] &= \mathbb{E}_{Z \sim N(0,1)}[\alpha^2 c^2 Z^2 \mathbb{1}_{\{Z > 0\}}] + \mathbb{E}_{Z \sim N(0,1)}[\beta^2 c^2 Z^2 \mathbb{1}_{\{Z \le 0\}}] \\
&= \frac{\alpha^2 + \beta^2}{2} \cdot \mathbb{E}_{Z \sim N(0,1)}[c^2 Z^2].
\end{aligned}
$$

Accordingly, for any positive semi-definite matrix $\Sigma \in \mathcal{S}$ and $i \in [n]$, we have

$$
\begin{aligned}
G(\Sigma)_{ii} &= \mathbb{E}_{h \sim N(\mathbf{0}_n, \Sigma)}[\sigma(h_i)^2] = \mathbb{E}_{Z \sim N(0,1)}\left[\sigma(\sqrt{\Sigma_{ii}}Z)^2\right] \\
&= \frac{\alpha^2 + \beta^2}{2} \cdot \mathbb{E}_{Z \sim N(0,1)}\left[\Sigma_{ii} Z^2\right] = \frac{\alpha^2 + \beta^2}{2} \cdot \Sigma_{ii}.
\end{aligned}
\tag{25}
$$

Combining (24) and (25), we get

$$\mathrm{tr}(\Sigma^{(l+1)}) \le \frac{\sigma_w^2 (\alpha^2 + \beta^2)}{2} \, \mathrm{tr}(\Sigma^{(l)}).$$

Thus, we have shown that $\{\mathrm{tr}(\Sigma^{(l)})\}_{l=1}^{\infty}$ is a decreasing sequence if $\sigma_w \le 2/(\alpha^2 + \beta^2)$. In addition, if $\sigma_w < 2/(\alpha^2 + \beta^2)$, we get

$$\mathrm{tr}(\Sigma^{(L)}) \le \left(\frac{\sigma_w^2 (\alpha^2 + \beta^2)}{2}\right)^{L-1} \mathrm{tr}(\Sigma^{(1)}).
\tag{26}$$

By Proposition D.1, we have

$$\operatorname{tr}(\Sigma^{(1)}) = \frac{\sigma_w^2}{d_0} \operatorname{tr}(\hat{A} X X^T \hat{A}) = \frac{\sigma_w^2}{d_0} \sum_{k=1}^{d_0} \operatorname{tr}(\hat{A} X_{:,k} X_{:,k}^T \hat{A}) = \frac{\sigma_w^2}{d_0} \sum_{k=1}^{d_0} \|\hat{A} X_{:,k}\|^2 \qquad (27)$$

Since all the eigenvalues of $\hat{A}$ belong to $(-1, 1]$ by Propositon C.1, we get

$$\text{RHS of (27)} \leq \frac{\sigma_w^2}{d_0} \sum_{k=1}^{d_0} \|X_{:,k}\|^2 = \frac{\sigma_w^2}{d_0} \operatorname{tr}(X X^T). \qquad (28)$$

Combining (26), (27), and (28), we have

$$\operatorname{tr}(\Sigma^{(L)}) \leq \frac{\sigma_w^2}{d_0} \cdot \left( \frac{\sigma_w^2(\alpha^2 + \beta^2)}{2} \right)^{L-1} \operatorname{tr}(X X^T).$$

Thus, the forward propagation metric at the $L$-th layer satisfies

$$\begin{aligned}
\mathbb{E}_{H \sim N(\mathbf{0}_n, \Sigma^{(L)})} \left[ \frac{\|H\|_{\mathrm{F}}^2}{\|X\|_{\mathrm{F}}^2} \right] &= \frac{C}{\operatorname{tr}(X X^T)} \cdot \mathbb{E}_{h \sim N(\mathbf{0}_n, \Sigma^{(L)})}[\|h\|^2] = \frac{C}{\operatorname{tr}(X X^T)} \operatorname{tr}(\Sigma^{(L)}) \\
&\leq \frac{C \sigma_w^2}{d_0} \cdot \left( \frac{\sigma_w^2(\alpha^2 + \beta^2)}{2} \right)^{L-1} \\
&= \frac{2C}{(\alpha^2 + \beta^2) d_0} \cdot \left( \frac{\sigma_w^2(\alpha^2 + \beta^2)}{2} \right)^{L}.
\end{aligned}$$

Then we have completed part 2 of this theorem. If $\sigma$ is ReLU activation function, i.e., $(1, 0)$-ReLU. If $\sigma < 2 = \frac{2}{1^2 + 0^2}$, we have

$$\mathbb{E}_{H \sim N(\mathbf{0}_n, \Sigma^{(L)})} \left[ \frac{\|H\|_{\mathrm{F}}^2}{\|X\|_{\mathrm{F}}^2} \right] = \frac{2C}{(1^2 + 0^2) d_0} \cdot \left( \frac{\sigma_w^2(1^2 + 0^2)}{2} \right)^L = \frac{2C}{d_0} \cdot \left( \frac{\sigma_w^2}{2} \right)^L,$$

which coincides with **part 2 in Theorem 3.1**.

Next, we will prove **part 1** of this theorem. Let's study the case when $\sigma_w^2 = 2/(\alpha^2 + \beta^2)$. Suppose that

$$\lim_{l \to \infty} \operatorname{tr}(\Sigma^{(l)}) = \delta_0.$$

If $\delta_0 = 0$, then we have completed the first part of this theorem by getting

$$\begin{aligned}
\lim_{L \to \infty} \mathbb{E}_{H \sim N(\mathbf{0}_n, \Sigma^{(L)})}[\|H\|_{\mathrm{F}}^2 / \|X\|_{\mathrm{F}}^2] &= \lim_{L \to \infty} \frac{C}{\|X\|_{\mathrm{F}}^2} \cdot \mathbb{E}_{h \sim N(\mathbf{0}_n, \Sigma^{(L)})}[\|h\|^2] \\
&= \frac{C}{\|X\|_{\mathrm{F}}^2} \cdot \lim_{L \to \infty} \operatorname{tr}(\Sigma^{(L)}) = 0.
\end{aligned}$$

Now we study the case when $\delta_0 > 0$. In order to show part 2 of the theorem, we only need to demonstrate that

$$\lim_{L \to \infty} \mathbb{E}_{H \sim N(\mathbf{0}_n, \Sigma^{(L)})} \left[ \frac{\operatorname{Dir}(H)}{\|H\|_{\mathrm{F}}^2} \right] = 0.$$

Given any fixed $\epsilon > 0$, we have

$$\begin{aligned}
&\mathbb{E}_{H \sim N(\mathbf{0}_n, \Sigma^{(L)})} \left[ \frac{\operatorname{Dir}(H)}{\|H\|_{\mathrm{F}}^2} \right] \\
&= \mathbb{E}_{H \sim N(\mathbf{0}_n, \Sigma^{(L)})} \left[ \frac{\operatorname{Dir}(H)}{\|H\|_{\mathrm{F}}^2} \mathbb{1}_{\{\|H\|_{\mathrm{F}} \geq \epsilon\}} \right] + \mathbb{E}_{H \sim N(\mathbf{0}_n, \Sigma^{(L)})} \left[ \frac{\operatorname{Dir}(H)}{\|H\|_{\mathrm{F}}^2} \mathbb{1}_{\{\|H\|_{\mathrm{F}} \leq \epsilon\}} \right].
\end{aligned} \qquad (29)$$

From Lemma C.2, it holds that $\operatorname{Dir}(H)/\|H\|_{\mathrm{F}}^2 \leq 2$, so we get

$$\begin{aligned}
\text{RHS of (29)} &\leq \frac{1}{\epsilon^2} \cdot \mathbb{E}_{H \sim N(\mathbf{0}_n, \Sigma^{(L)})} \left[ \operatorname{Dir}(H) \mathbb{1}_{\{\|H\|_{\mathrm{F}} \geq \epsilon\}} \right] + 2 \cdot \mathbb{P}_{H \sim N(\mathbf{0}_n, \Sigma^{(L)})} \left[ \|H\|_{\mathrm{F}} \leq \epsilon \right] \\
&\leq \frac{1}{\epsilon^2} \cdot \mathbb{E}_{H \sim N(\mathbf{0}_n, \Sigma^{(L)})} [\operatorname{Dir}(H)] + 2 \cdot \mathbb{P}_{H \sim N(\mathbf{0}_n, \Sigma^{(L)})} \left[ \|H\|_{\mathrm{F}} \leq \epsilon \right].
\end{aligned} \qquad (30)$$

For any $L \geq 1$, there exists $i \in [n]$, such that $\Sigma_{ii}^{(L)} \geq \text{tr}(\Sigma^{(L)})/n$. Then for any $n \times C$ random matrix $H \sim N(\mathbf{0}_n, \Sigma^{(L)})$, we have $H_{i,1} \sim N(0, \Sigma_{ii}^{(L)})$. For this reason, we have

$$
\begin{aligned}
\mathbb{P}_{H \sim N(\mathbf{0}_n, \Sigma^{(L)})} \left[ \|H\|_{\text{F}} \leq \epsilon \right] &\leq \mathbb{P}_{H \sim N(\mathbf{0}_n, \Sigma^{(L)})} \left[ |H_{i,1}| \leq \epsilon \right] = \mathbb{P}_{Z \sim N(0,1)} \left[ |Z| \leq \frac{\epsilon}{\sqrt{\Sigma_{ii}}} \right] \\
&\leq \mathbb{P}_{Z \sim N(0,1)} \left[ |Z| \leq \epsilon \cdot \sqrt{\frac{n}{\text{tr}(\Sigma^{(L)})}} \right] \\
&= 2\Phi \left( \epsilon \cdot \sqrt{\frac{n}{\text{tr}(\Sigma^{(L)})}} \right) - 1,
\end{aligned}
\tag{31}
$$

where $\Phi(x) = \mathbb{P}_{Z \sim N(0,1)}[Z \leq x]$ denotes the cumulative distribution function of the standard normal distribution $N(0, 1)$.

Combining (29), (30), and (31), we get

$$
\mathbb{E}_{H \sim N(\mathbf{0}_n, \Sigma^{(L)})} \left[ \frac{\text{Dir}(H)}{\|H\|_{\text{F}}^2} \right] \leq \frac{1}{\epsilon^2} \cdot \mathbb{E}_{H \sim N(\mathbf{0}_n, \Sigma^{(L)})} \left[ \text{Dir}(H) \right] + 4\Phi \left( \epsilon \cdot \sqrt{\frac{n}{\text{tr}(\Sigma^{(L)})}} \right) - 2,
$$

for any $L \geq 1$.

Since

$$
\sigma_w^2 = \frac{2}{\alpha^2 + \beta^2} < \frac{2}{\lambda^2(\alpha^2 + \beta^2)},
$$

by Lemma D.12, we have

$$
\lim_{L \to \infty} \mathbb{E}_{H \sim N(\mathbf{0}_n, \Sigma^{(L)})}[\text{Dir}(H)] = 0.
$$

We let $L \to \infty$ in (29) and get

$$
\begin{aligned}
&\limsup_{L \to \infty} \mathbb{E}_{H \sim N(\mathbf{0}_n, \Sigma^{(L)})} \left[ \frac{\text{Dir}(H)}{\|H\|_{\text{F}}^2} \right] \\
&\leq \frac{1}{\epsilon^2} \cdot \limsup_{L \to \infty} \mathbb{E}_{H \sim N(\mathbf{0}_n, \Sigma^{(L)})} \left[ \text{Dir}(H) \right] + 4 \cdot \limsup_{L \to \infty} \Phi \left( \epsilon \cdot \sqrt{\frac{n}{\text{tr}(\Sigma^{(L)})}} \right) - 2 \\
&= \frac{1}{\epsilon^2} \cdot 0 + 4\Phi \left( \epsilon \cdot \sqrt{\frac{n}{\delta_0}} \right) - 2 = 4\Phi \left( \epsilon \cdot \sqrt{\frac{n}{\delta_0}} \right) - 2.
\end{aligned}
\tag{32}
$$

Notice that the left hand side of (32) is independent of the choice of $\epsilon$. Since $\Phi$ is a continuous map, we let $\epsilon \to 0^+$ and get

$$
\limsup_{L \to \infty} \mathbb{E}_{H \sim N(\mathbf{0}_n, \Sigma^{(L)})} \left[ \frac{\text{Dir}(H)}{\|H\|_{\text{F}}^2} \right] \leq 4\Phi(0) - 2 = 0.
$$

Therefore, we have

$$
\lim_{L \to \infty} \mathbb{E}_{H \sim N(\mathbf{0}_n, \Sigma^{(L)})} \left[ \frac{\text{Dir}(H)}{\|H\|_{\text{F}}^2} \right] = 0.
$$

$\square$

### D.4 PROOF OF THEOREM 3.2 (SIGNAL PROPAGATION ON RELU-ACTIVATED VANILLA GCN)

**Theorem D.13** (The generalized version of Theorem 3.2). *Under the NNGP correspondence approximation, the graph embedding variation metric $\mathbf{M}_{GEV}^{(L)}(\sigma_w^2) = \mathbb{E}_{H \sim N(\mathbf{0}_n, \Sigma^{(L)})}[\text{Dir}(H)/\|H\|_{\text{F}}^2]$ is independent of the choice of $\sigma_w^2$.*

*Proof of Theorem D.13 (the generalized version of Theorem 3.2).* Under the NNGP correspondence approximation, we only need to prove that

$$
\frac{\Sigma^{(l)}(\sigma_w^2)}{\sigma_w^{2l}} = \frac{\Sigma^{(l)}(\tilde{\sigma}_w^2)}{\tilde{\sigma}_w^{2l}}, \quad \text{for any } l \geq 1 \text{ and } \sigma_w^2, \tilde{\sigma}_w^2 > 0.
\tag{33}
$$

If (33) holds, then $H \sim N(\mathbf{0}_n, \Sigma^{(L)}(\sigma_w^2))$ implies $\tilde{\sigma}_w^L H / \sigma_w^L \sim N(\mathbf{0}_n, \Sigma^{(L)}(\tilde{\sigma}_w^2))$. In this way, we have

$$\mathbb{E}_{H \sim N(\mathbf{0}_n, \Sigma^{(L)}(\sigma_w^2))} \left[ \frac{\mathrm{Dir}(H)}{\|H\|_F^2} \right] = \mathbb{E}_{H \sim N(\mathbf{0}_n, \Sigma^{(L)}(\sigma_w^2))} \left[ \frac{\mathrm{Dir}(\tilde{\sigma}_w^L H / \sigma_w^L)}{\|\tilde{\sigma}_w^L H / \sigma_w^L\|_F^2} \right]$$

$$= \mathbb{E}_{H \sim N(\mathbf{0}_n, \Sigma^{(L)}(\tilde{\sigma}_w^2))} \left[ \frac{\mathrm{Dir}(H)}{\|H\|_F^2} \right].$$

Now we prove (33) by mathematical induction. When $l = 1$, by Proposition D.1, we have

$$\frac{\Sigma^{(1)}(\sigma_w^2)}{\sigma_w^2} = \frac{1}{d_0} \hat{A} X X^T \hat{A} = \frac{\Sigma^{(1)}(\tilde{\sigma}_w^2)}{\tilde{\sigma}_w^2}, \quad \text{for any } \sigma_w^2, \tilde{\sigma}_w^2 > 0.$$

If (33) holds for $L$, we look at the case for $L + 1$. Since the activation $\sigma$ is $(\alpha, \beta)$-ReLU, for any $c \in \mathbb{R}_+$, we have $\sigma(cx) = c\sigma(x)$. Recalling the definition of $G$ in Proposition D.1, for any positive semi-definite matrix $\Sigma \in \mathcal{S}$, we have

$$G(c^2 \Sigma)_{ij} = \mathbb{E}_{h \sim N(\mathbf{0}_n, c^2 \Sigma)}[\sigma(h_i) \cdot \sigma(h_j)] = \mathbb{E}_{h \sim N(\mathbf{0}_n, \Sigma)}[\sigma(ch_i) \cdot \sigma(ch_j)]$$

$$= c^2 \mathbb{E}_{h \sim N(\mathbf{0}_n, \Sigma)}[\sigma(h_i) \cdot \sigma(h_j)] = c^2 G(\Sigma)_{ij},$$

for any $i, j \in [n]$ and $c \in \mathbb{R}_+$. Thus, by Proposition D.1, we have

$$\left( \frac{\tilde{\sigma}_w^2}{\sigma_w^2} \right)^{L+1} \cdot \Sigma^{(L+1)}(\sigma_w^2) \overset{(a)}{=} \left( \frac{\tilde{\sigma}_w^2}{\sigma_w^2} \right)^{L+1} \cdot \sigma_w^2 \hat{A} G \left( \Sigma^{(L)}(\sigma_w^2) \right) \hat{A}$$

$$= \tilde{\sigma}_w^2 \cdot \left( \frac{\tilde{\sigma}_w^2}{\sigma_w^2} \right)^L \cdot \hat{A} G \left( \Sigma^{(L)}(\sigma_w^2) \right) \hat{A}$$

$$\overset{(b)}{=} \tilde{\sigma}_w^2 \cdot \hat{A} G \left( \Sigma^{(L)}(\tilde{\sigma}_w^2) \right)$$

$$\overset{(c)}{=} \Sigma^{(L+1)}(\tilde{\sigma}_w^2),$$

where $(a)$ and $(c)$ are due to Proposition D.1 and $(b)$ are from the induction hypothesis.

Therefore, (33) holds for all $L \geq 1$ and we have completed the proof. $\qquad \square$

## D.5 Signal propagation on tanh-activated vanilla GCN

**Theorem D.14.** *Under the NNGP correspondence approximation, when the activation function $\sigma$ is tanh, we have*

1. *When $\sigma_w^2 = 1$, we have $\lim_{L \to \infty} \mathbf{M}_{FSP}^{(L)}(\sigma_w^2) = \lim_{L \to \infty} \mathbb{E}_{H \sim N(\mathbf{0}_n, \Sigma^{(L)})}[\|H\|_F^2 / \|X\|_F^2] = 0$.*

2. *When $\sigma_w^2 < 1$, we have $\mathbf{M}_{FSP}^{(L)}(\sigma_w^2) = \mathbb{E}_{H \sim N(\mathbf{0}_n, \Sigma^{(L)})}[\|H\|_F^2 / \|X\|_F^2] \leq \frac{C}{d_0} \cdot \sigma_w^{2L}$ for any $L \geq 1$.*

**Lemma D.15.** *The collection of positive semi-definite matrices $\mathcal{S}$ defined by (9) is a closed subset of $\mathbb{R}^{n \times n}$.*

*Proof.* We only need to show that given any convergent sequence $\{Q^{(k)}\}_{k=1}^\infty \subset \mathcal{S}$, its limit also belongs to $\mathcal{S}$. Suppose that

$$\lim_{k \to \infty} Q^{(k)} = Q^*.$$

Since all $\{Q^{(k)}\}_{k=1}^\infty$ are positive semi-definite matrices, so given any $x \in \mathbb{R}^{n \times 1}$, we have

$$x^T Q^{(k)} x \geq 0, \quad \text{for all } k \in \mathbb{N}.$$

Then we get

$$x^T Q^* x = \lim_{k \to \infty} x^T Q^{(k)} x \geq 0.$$

Thus, $Q^*$ also belongs to $\mathcal{S}$. $\qquad \square$

**Lemma D.16.** *When the activation function $\sigma$ is tanh, i.e., $\sigma(x) = (e^x - e^{-x})/(e^x + e^{-x})$, then we have $|\sigma(x)| \leq |x|$ for any $x \in \mathbb{R}$. Moreover, the equality holds if and only if $x = 0$.*

*Proof.* It is easy to verify that $\sigma(0) = 0$. Given any $x \geq 0$, we have

$$\sigma(-x) = \frac{e^{-x} - e^x}{e^{-x} + e^x} = -\frac{e^x - e^{-x}}{e^{-x} + e^x} = -\sigma(x).$$

For this reason, we only need to prove that $|\sigma(x)| < |x|$ for any $x > 0$. In the following part, we will show that $0 < \sigma(x) < x$ when $x > 0$.

We define $f(x) := \sigma(x) - x$ for any $x \geq 0$. Let's consider the derivative of $f$:

$$
\begin{aligned}
f'(x) &= \frac{d}{dx} \left( \frac{e^x - e^{-x}}{e^x + e^{-x}} - x \right) \\
&= \frac{1}{(e^x + e^{-x})^2} \left[ (e^x + e^{-x}) \cdot \frac{d}{dx}(e^x - e^{-x}) - (e^x - e^{-x}) \cdot \frac{d}{dx}(e^x + e^{-x}) \right] - 1 \\
&= \frac{(e^x + e^{-x})^2 - (e^x - e^{-x})^2}{(e^x + e^{-x})^2} - 1 \\
&= \frac{-(e^x - e^{-x})^2}{(e^x + e^{-x})^2}.
\end{aligned}
$$

Then if $x > 0$, we have $f'(x) < 0$; if $x = 0$, we have $f'(x) = 0$. Thus, $f(x) = \sigma(x) - x$ is a strictly decreasing function in $[0, +\infty)$. Since $f(0) = \sigma(0) - 0 = 0$, we have

$$f(x) = \sigma(x) - x < 0, \quad \text{for any } x > 0.$$

Since $0 < e^x - e^{-x} < e^x + e^{-x}$ for any $x > 0$, it holds that

$$\sigma(x) = (e^x - e^{-x})/(e^x + e^{-x}) > 0, \quad \text{for any } x > 0.$$

Therefore, we get that $0 < \sigma(x) < x$ for any $x > 0$ and have completed the proof of this lemma. $\quad\square$

Now it is time for Theorem D.14.

*Proof of Theorem D.14.* First of all, we will prove **part 2** of this theorem. For any positive semi-definite matrix $\Sigma \in \mathcal{S}$, we have

$$\mathbb{E}_{h \sim N(\mathbf{0}_n, \Sigma)}[\|h\|^2] = \mathbb{E}_{h \sim N(\mathbf{0}_n, \Sigma)}[\text{tr}(h^T h)] = \mathbb{E}_{h \sim N(\mathbf{0}_n, \Sigma)}[\text{tr}(hh^T)] = \text{tr}(\Sigma).$$

For this reason, we only need to focus on $\{\text{tr}(\Sigma^{(l)})\}_{l=1}^{\infty}$ in the following proof.

We will show that $\{\text{tr}(\Sigma^{(l)})\}_{l=1}^{\infty}$ is a decreasing sequence if $\sigma_w \leq 1$. By Lemma D.5, we have

$$\text{tr}(\Sigma^{(l+1)}) \leq \sigma_w^2 \, \text{tr}(G(\Sigma^{(l)})). \tag{34}$$

By Lemma D.16, we have $|\sigma(x)| \leq |x|$ for any $x \in \mathbb{R}$. Moreover, the equality holds if and only if $x = 0$. For this reason, given any positive semi-definite matrix $\Sigma \in \mathcal{S}$, we have

$$
\begin{aligned}
\text{tr}(G(\Sigma)) &= \sum_{i=1}^{n} \mathbb{E}_{h \sim N(\mathbf{0}_n, \Sigma)}[\sigma(h_i)^2] = \sum_{i=1}^{n} \mathbb{E}_{Z \sim N(0,1)} \left[ \sigma(\sqrt{\Sigma_{ii}} Z)^2 \right] \\
&\leq \sum_{i=1}^{n} \mathbb{E}_{Z \sim N(0,1)} \left[ (\sqrt{\Sigma_{ii}} Z)^2 \right] = \sum_{i=1}^{n} \mathbb{E}_{h \sim N(\mathbf{0}_n, \Sigma)}[h_i^2] = \text{tr}(\Sigma),
\end{aligned}
\tag{35}
$$

and the inequality becomes an equality if and only if $\sqrt{\Sigma_{ii}} Z = 0$ holds $\mathbb{P}$-a.s. for all $i \in [n]$. Since $Z \sim N(0, 1)$ follows a standard normal distribution, it is equivalent to $\Sigma_{ii} = 0$ for all $i \in [n]$, i.e., $\text{tr}(\Sigma) = 0$.

Combining (34) and (35), we get

$$\text{tr}(\Sigma^{(l+1)}) \leq \sigma_w^2 \, \text{tr}(\Sigma^{(l)}).$$

Thus, we have shown that $\{\text{tr}(\Sigma^{(l)})\}_{l=1}^{\infty}$ is a decreasing sequence if $\sigma_w \leq 1$. In addition, if $\sigma_w < 1$, we get

$$\text{tr}(\Sigma^{(L)}) \leq \sigma_w^{2(L-1)} \text{tr}(\Sigma^{(1)}). \tag{36}$$

Analogous to the proof of part 2 in Theorem D.7 for ReLU-activated model, by Proposition D.1 and Proposition C.1, we have

$$\text{tr}(\Sigma^{(1)}) = \frac{\sigma_w^2}{d_0} \text{tr}(\hat{A} X X^T \hat{A}) = \frac{\sigma_w^2}{d_0} \sum_{k=1}^{d_0} \text{tr}(\hat{A} X_{:,k} X_{:,k}^T \hat{A})$$
$$= \frac{\sigma_w^2}{d_0} \sum_{k=1}^{d_0} \|\hat{A} X_{:,k}\|^2 \leq \frac{\sigma_w^2}{d_0} \sum_{k=1}^{d_0} \|X_{:,k}\|^2 = \frac{\sigma_w^2}{d_0} \|X\|_F^2. \tag{37}$$

Combining (36) and (37), we have

$$\text{tr}(\Sigma^{(1)}) \leq \frac{\sigma_w^{2L}}{d_0} \|X\|_F^2.$$

Then we have completed part 2 of the theorem by getting

$$\mathbb{E}_{H \sim N(\mathbf{0}_n, \Sigma^{(L)})} \left[ \frac{\|H\|_F^2}{\|X\|_F^2} \right] = \frac{C}{\|X\|_F^2} \mathbb{E}_{h \sim N(\mathbf{0}_n, \Sigma^{(L)})}[\|h\|^2] = \frac{C}{\|X\|_F^2} \text{tr}(\Sigma^{(L)})$$
$$\leq \frac{C}{\|X\|_F^2} \cdot \frac{\sigma_w^{2L}}{d_0} \cdot \|X\|_F^2 \leq \frac{C}{d_0} \cdot \sigma_w^{2L}.$$

Next, we will prove **part 1** of this theorem. Let's study the case when $\sigma_w = 1$.

Since $\Sigma^{(l)}$ is a positive semi-definite matrix for any $l \in \mathbb{N}$, we have

$$|\Sigma_{ij}^{(l)}|^2 \leq \Sigma_{ii}^{(l)} \Sigma_{jj}^{(l)} \leq \text{tr}(\Sigma^{(l)})^2 \leq \text{tr}(\Sigma^{(1)})^2, \quad \text{for all } i, j \in [n].$$

Taking the summation of both sides w.r.t. $i$ and $j$, we get

$$\|\Sigma^{(l)}\|_F^2 = \sum_{i,j=1}^{n} |\Sigma_{ij}^{(l)}|^2 \leq n^2 \text{tr}(\Sigma^{(1)})^2 < \infty.$$

Thus, the matrix sequence $\{\Sigma^{(l)}\}_{l=1}^{\infty}$ lies in

$$\mathcal{S}' = \mathcal{S} \cap \{\Sigma \in \mathbb{R}^{n \times n} : \|\Sigma\|_F \leq n \text{tr}(\Sigma^{(1)})\}.$$

By Lemma D.15, $\mathcal{S}'$ is a bounded and closed subset, i.e., a compact subset, of $\mathbb{R}^{n \times n}$. By the Bolzano–Weierstrass theorem, there exists a subsequence $\{\Sigma^{(l_k)}\}_{k=1}^{\infty}$ of $\{\Sigma^{(l)}\}_{l=1}^{\infty}$ and $\Sigma^* \in \mathcal{S}'$ such that

$$\lim_{k \to \infty} \Sigma^{(l_k)} = \Sigma^*.$$

Recalling (34) and that $\{\text{tr}(\Sigma^{(l)})\}_{l=1}^{\infty}$ is a decreasing sequence, we have

$$\text{tr}(\Sigma^{(l_{k+1})}) \leq \text{tr}(\Sigma^{(l_k+1)}) \leq \text{tr}(G(\Sigma^{(l_k)})).$$

Since $G$ is a continuous function, we let $k \to \infty$ and get

$$\text{tr}(\Sigma^*) = \lim_{k \to \infty} \text{tr}(\Sigma^{(l_{k+1})}) \leq \lim_{k \to \infty} \text{tr}(G(\Sigma^{(l_k)})) = \text{tr}(G(\Sigma^*)).$$

According to (35), we have

$$\text{tr}(G(\Sigma^*)) = \text{tr}(\Sigma^*).$$

This implies $\text{tr}(\Sigma^*) = 0$ by (35).

Then, since $\{\text{tr}(\Sigma^{(l)})\}_{l=1}^{\infty}$ is a decreasing sequence, we have

$$\lim_{l \to \infty} \mathbb{E}_{h \sim N(\mathbf{0}_n, \Sigma^{(l)})}[\|h\|^2] = \lim_{l \to \infty} \text{tr}(\Sigma^{(l)}) = \lim_{k \to \infty} \text{tr}(\Sigma^{(l_k)}) = \text{tr}(\Sigma^*) = 0.$$

Consequently, we have

$$\lim_{L \to \infty} \mathbb{E}_{H \sim N(\mathbf{0}_n, \Sigma^{(L)})} \left[ \frac{\|H\|_F^2}{\|X\|_F^2} \right] = \frac{C}{\|X\|_F^2} \lim_{L \to \infty} \mathbb{E}_{h \sim N(\mathbf{0}_n, \Sigma^{(L)})}[\|h\|^2] = 0.$$

$\square$

# E  SIGNAL PROPAGATION THEORY FOR LINEAR RESGCN

## E.1  NNGP CORRESPONDENCE FOR LINEAR RESGCN

**Proposition E.1** (NNGP correspondence for linear ResGCN)**.** *As the width of the hidden layers* $d \to \infty$, *the l-th layer's pre-activation embedding channels* $\{H^{(l)}_{:,k}\}_{k \in [d]}$ *converge to i.i.d. Gaussian random variables* $N(\mathbf{0}_n, \tilde{\Sigma}^{(l)})$ *in distribution. The covariance matrices are*

$$\tilde{\Sigma}^{(1)} = \frac{\sigma_w^4}{d_0} \hat{A} X X^T \hat{A},$$
$$\tilde{\Sigma}^{(l+1)} = \alpha^2 \sigma_w^2 \hat{A} \tilde{\Sigma}^{(l)} \hat{A} + \beta^2 \tilde{\Sigma}^{(l)}. \tag{38}$$

*Moreover, as* $d \to \infty$, *the l-th layer's post-activation embedding channels* $\{X^{(l)}_{:,k}\}_{k \in [d]}$ *converge to i.i.d. random variables in distribution. The random variables have mean* $\mathbf{0}_n$ *and their covariance matrices* $\Phi^{(l)}$, *which satisfy*

$$\Phi^{(0)} = \frac{\sigma_w^2}{d_0} X X^T,$$
$$\Phi^{(l)} = \alpha^2 \sigma_w^2 \hat{A} \Phi^{(l-1)} \hat{A} + \beta^2 \Phi^{(l-1)}. \tag{39}$$

*Proof of Proposition E.1.* For $\Phi^{(l)}, \tilde{\Sigma}^{(l+1)}$ defined by (39) and (38), it is easy to show that $\tilde{\Sigma}^{(l+1)} = \sigma_w^2 \hat{A} \Phi^{(l)} \hat{A}$. Similar to the proof of Proposition D.1, We will prove this proposition by mathematical induction.

**Base case.** When $l = 0$, the $k$-th channel of $X^{(0)}$ is

$$X^{(0)}_{:,k} = X W^{(0)}_{:,k} + \mathbf{1}_n \cdot b^{(0)}_k = X W^{(0)}_{:,k}. \tag{40}$$

According to our initialization, the weights $\{W^{(0)}_{:,k}\}_{k \in [d]}$ are i.i.d. random variables, so $\{X^{(0)}_{:,k}\}_{k \in [d]}$ are also i.i.d. random variables. Taking the expectation of (40), we get

$$\mathbb{E}[X^{(0)}_{:,k}] = X \cdot \mathbb{E}[W^{(0)}_{:,k}] = \mathbf{0}_n.$$

Calculating the covariance matrix of (40), we have

$$\mathrm{Cov}[X^{(0)}_{:,k}] = \mathbb{E}[X^{(0)}_{:,k} \cdot X^{(0)T}_{:,k}] = \mathbb{E}[X W^{(0)}_{:,k} W^{(0)T}_{:,k} X^T]$$
$$= X \cdot \mathbb{E}[W^{(0)}_{:,k} W^{(0)T}_{:,k}] \cdot X^T = X \left( \frac{\sigma_w^2}{d_0} \cdot I_{d_0} \right) X^T$$
$$= \frac{\sigma_w^2}{d_0} X X^T.$$

Thus, if we let $\Phi^{(0)} = \sigma_w^2 X X^T / d_0$, then we have $\{X^{(0)}_{:,k}\}_{k \in [d]}$ are i.i.d. with mean $\mathbf{0}_n$ and covariance matrix $\Phi^{(0)}$.

Now we study the pre-activation embedding $H^{(1)}$. Since the bias term $b^{(1)}$ is initialized to be $\mathbf{0}_d$, we have

$$H^{(1)} = \hat{A} X^{(0)} W^{(1)} + \mathbf{1}_n \cdot b^{(1)} = \hat{A} X^{(0)} W^{(1)}.$$

Similar to the proof of Proposition D.1 for vanilla GCN, we vectorize the equation and get

$$\mathrm{vec}(H^{(1)}) = \sum_{k=1}^{d} \mathrm{vec} \left( \underbrace{[\hat{A} X^{(0)}_{:,k}]}_{n \times 1} \cdot \underbrace{W^{(1)}_{k,:}}_{1 \times d} \right).$$

For brevity, we define

$$\omega^{(1)}_{kk'} := \sqrt{d} \cdot W^{(1)}_{kk'}, \quad \text{for all } k, k' \in [d]$$

and

$$Z^{(1)}_k := \mathrm{vec} \left( [\hat{A} X^{(0)}_{:,k}] \cdot w^{(l+1)}_{k,:} \right), \text{for all } k \in [d].$$

Then we get that $\{\omega_{kk'}^{(1)}\}_{k,k' \in [d]}$ are i.i.d. with mean $0$ and variance $\sigma_w^2$, and

$$\mathrm{vec}(H^{(1)}) = \frac{1}{\sqrt{d}} \sum_{k=1}^{d} Z_k^{(1)}.$$

Analogous to the proof of Proposition D.1, $\{Z_k^{(1)}\}_{k \in [d]}$ are i.i.d., $\mathbb{E}[Z_1^{(1)}] = \mathbf{0}_{nd}$, and

$$\begin{aligned}
\mathrm{Cov}[Z_1^{(1)}] &= \mathbb{E}\left[\omega_{1,:}^{(1)T} \omega_{1,:}^{(1)}\right] \otimes \left\{\hat{A} \cdot \mathbb{E}\left[X_{:,1}^{(0)} X_{:,1}^{(0)T}\right] \cdot \hat{A}\right\} \\
&= \sigma_w^2 I_d \otimes \hat{A}\Phi^{(0)}\hat{A} \\
&= I_d \otimes \sigma_w^2 \hat{A}\Phi^{(0)}\hat{A}.
\end{aligned}$$

Since $\tilde{\Sigma}^{(1)} = \sigma_w^4 \hat{A}XX^T\hat{A}/d_0 = \sigma_w^2 \hat{A}\Phi^{(0)}\hat{A}$, applying the central limit theorem, $\mathrm{vec}(H^{(1)})$ converges to a Gaussian random variable $N(\mathbf{0}_{nd}, I_d \otimes \tilde{\Sigma}^{(1)})$ as $d \to \infty$. Consequently, $\{H_{:,k}^{(1)}\}$ converge to i.i.d. Gaussian random variables $N(\mathbf{0}_n, \tilde{\Sigma}^{(1)})$ in distribution.

**Induction step.** Suppose that $\{X_{:,k}^{(l-1)}\}_{k \in [d]}$ converge to i.i.d. random variables with mean $\mathbf{0}_n$ and covariance matrix $\Phi^{(l-1)}$ in distribution. Suppose that $\{H_{:,k}^{(l)}\}_{k \in [d]}$ converge to i.i.d. Gaussian random variables $N(\mathbf{0}_n, \tilde{\Sigma}^{(l)})$ in distribution. Now we look at $X^{(l)}$ first.

For the linear ResGCN at initialization, the post-activation embeddings satisfy

$$X^{(l)} = \alpha H^{(l)} + \beta X^{(l-1)} = \alpha \hat{A}X^{(l-1)}W^{(l)} + \beta X^{(l-1)}$$

We take any $k$-th channel $X_{:,k}^{(l)}$ of $X^{(l)}$:

$$\begin{aligned}
X_{:,k}^{(l)} &= \alpha H_{:,k}^{(l)} + \beta X_{:,k}^{(l-1)} \\
&= \alpha \hat{A}X^{(l-1)}W_{:,k}^{(l)} + \beta X_{:,k}^{(l-1)} \\
&= \alpha \left(\sum_{k'=1}^{d} \hat{A}X_{:,k'}^{(l-1)} W_{k'k}^{(l)}\right) + \beta X_{:,k}^{(l-1)} \\
&= \underbrace{\frac{\alpha}{\sqrt{d}}\left(\sum_{k'=1}^{d} \hat{A}X_{:,k'}^{(l-1)} \omega_{k'k}^{(l)}\right)}_{(i)} + \underbrace{\beta X_{:,k}^{(l-1)}}_{(ii)},
\end{aligned}$$

where $\omega_{k'k}^{(l)} := W_{k'k}^{(l)}/\sqrt{d}$ has mean $0$ and variance $\sigma_w^2$, which does not rely on $d$. By the induction hypothesis, $X_{:,k'}^{(l-1)}$ and $X_{:,k}^{(l-1)}$ are independent when $k' \neq k$. Then $(ii)$ is independent of the $k'$-th term $\alpha \hat{A}X_{:,k'}^{(l-1)} \omega_{k'k}^{(l)}/\sqrt{d}$ in $(i)$ when $k' \neq k$. We notice that the correlation between $(i)$'s $k$-th term $\alpha \hat{A}X_{:,k}^{(l-1)} \omega_{kk}^{(l)}/\sqrt{d}$ and $\beta X_{:,k}^{(l-1)}$ goes to $0$ as $d \to \infty$. Thus, we get that $(i)$ and $(ii)$ are asymptotically independent, the expectation

$$\mathbb{E}[X_{:,k}^{(l)}] = \alpha \mathbb{E}[H_{:,k}^{(l)}] + \beta \mathbb{E}[X_{:,k}^{(l-1)}] = \mathbf{0}_n,$$

and the covariance matrix

$$\begin{aligned}
\mathrm{Cov}[X_{:,k}^{(l)}] &= \alpha^2 \mathrm{Cov}[H_{:,k}^{(l)}] + \beta^2 \mathrm{Cov}[X_{:,k}^{(l-1)}] = \alpha^2 \tilde{\Sigma}^{(l)} + \beta^2 \Phi^{(l-1)} \\
&= \alpha^2 \sigma_w^2 \hat{A}\Phi^{(l-1)}\hat{A} + \beta^2 \Phi^{(l-1)} \\
&= \Phi^{(l)}.
\end{aligned}$$

By the induction hypothesis, $\{H_{:,k}^{(l)}\}$ are i.i.d. and $\{X_{:,k}^{(l-1)}\}$ are i.i.d. as $d \to \infty$, so $\{X_{:,k}^{(l)}\}$ are i.i.d..

Next, we look at the pre-activation embedding $H^{(l+1)}$. We have

$$H^{(l+1)} = \hat{A}X^{(l)}W^{(l+1)} + \mathbf{1}_n \cdot b^{(l+1)} = \hat{A}X^{(l)}W^{(l+1)}.$$

We also vectorize it and get

$$\text{vec}(H^{(l+1)}) = \sum_{k=1}^{d} \text{vec}([\hat{A}X_{:,k}^{(l)}] \cdot W_{k,:}^{(l+1)}).$$

Analogous to the proof of base case (or the proof of Proposition D.1), we can conclude that $\{H_{:,k}^{(l+1)}\}$ converge i.i.d. to $N(\mathbf{0}_n, \sigma_w^2 \hat{A}\Phi^{(l)}\hat{A})$, i.e. $N(\mathbf{0}_n, \tilde{\Sigma}^{(l+1)})$.

**Conclusion.** By the principle of mathematical induction, we have proven this proposition. $\qquad\square$

### E.2 PROOF OF THEOREM 3.3 (SIGNAL PROPAGATION ON LINEAR RESGCN)

**Theorem E.2.** *Suppose that there exists an eigenvector $u$ of $\hat{A}$ corresponding to the eigenvalue $1$, such that the input feature $X \in \mathbb{R}^{n \times d_0}$ satisfies $X^T u \neq \mathbf{0}_{d_0 \times 1}$. Under the NNGP correspondence approximation for linear ResGCN, if $\alpha^2 \sigma_w^2 + \beta^2 > 1$ and $\alpha \neq 0$, then we have*

*1.* $\lim_{L \to \infty} \mathbf{M}_{FSP}^{(L)}(\sigma_w^2) = \lim_{L \to \infty} \mathbb{E}_{H^{(L)} \sim N(\mathbf{0}_n, \tilde{\Sigma}^{(L)})}[\|H^{(L)}\|_F^2 / \|X\|_F^2] = +\infty.$

*2.* $\lim_{L \to \infty} \mathbf{M}_{GEV}^{(L)}(\sigma_w^2) = \lim_{L \to \infty} \mathbb{E}_{H^{(L)} \sim N(\mathbf{0}_n, \tilde{\Sigma}^{(L)})}[\text{Dir}(H^{(L)}) / \|H^{(L)}\|_F^2] = 0.$

*Proof of part* $1$ *in Theorem 3.3.* For any positive semi-definite matrix $\Sigma \in \mathcal{S}$, we have

$$\mathbb{E}_{h \sim N(\mathbf{0}_n, \Sigma)}[\|h\|^2] = \mathbb{E}_{h \sim N(\mathbf{0}_n, \Sigma)}[\text{tr}(h^T h)] = \mathbb{E}_{h \sim N(\mathbf{0}_n, \Sigma)}[\text{tr}(hh^T)] = \text{tr}(\Sigma).$$

Recalling the NNGP correspondence formula for linear ResGCN (38) in Proposition E.1, we have

$$\begin{aligned}
\tilde{\Sigma}^{(1)} &= \frac{\sigma_w^4}{d_0} \hat{A}XX^T\hat{A}, \\
\tilde{\Sigma}^{(l+1)} &= \sigma_w^2 \alpha^2 \hat{A}\tilde{\Sigma}^{(l)}\hat{A} + \beta^2 \tilde{\Sigma}^{(l)}.
\end{aligned} \tag{41}$$

By Proposition C.1, we can assume that $A = U\Lambda U^T$, where $\Lambda = \text{diag}(\lambda_1, \dots, \lambda_n)$ with $1 = \lambda_1 \geq \cdots \geq \lambda_n > -1$ and $U \in \mathbb{R}^{n \times n}$ is an orthogonal matrix, i.e., $UU^T = U^TU = I_n$. Then from (41), we get

$$\begin{aligned}
U^T\tilde{\Sigma}^{(l+1)}U &= \sigma_w^2 \alpha^2 \cdot U^T\hat{A}\tilde{\Sigma}^{(l)}\hat{A}U + \beta^2 \cdot U^T\tilde{\Sigma}^{(l)}U \\
&= \sigma_w^2 \alpha^2 \cdot \Lambda U^T\tilde{\Sigma}^{(l)}U\Lambda + \beta^2 \cdot U^T\tilde{\Sigma}^{(l)}U.
\end{aligned} \tag{42}$$

So for any $i \in [n]$ and $l \in \mathbb{N}$, we have

$$\begin{aligned}
(U^T\tilde{\Sigma}^{(l+1)}U)_{ii} &= \sigma_w^2 \alpha^2 \cdot \lambda_i (U^T\tilde{\Sigma}^{(l)}U)_{ii}\lambda_i + \beta^2 (U^T\tilde{\Sigma}^{(l)}U)_{ii} \\
&= (\alpha^2 \lambda_i^2 \sigma_w^2 + \beta^2) \cdot (U^T\tilde{\Sigma}^{(l)}U)_{ii}.
\end{aligned}$$

Thus, for any $i \in [n]$ and $L \in \mathbb{N}$, we have

$$(U^T\tilde{\Sigma}^{(L)}U)_{ii} = (\alpha^2 \lambda_i^2 \sigma_w^2 + \beta^2)^{L-1} \cdot (U^T\tilde{\Sigma}^{(1)}U)_{ii}.$$

According to the assumption on input feature $X$, there exists an eigenvector $u$ of $\hat{A}$ corresponding to the eigenvalue $1$, such that $X^T u \neq \mathbf{0}_{d_0 \times 1}$. Suppose that $u_1, u_2, \dots, u_n \in \mathbb{R}^{n \times 1}$ are the columns of $U$, then there exists $i \in [n]$ such that $X^T u_i \neq 0$. Otherwise, suppose that $u = \sum_{j=1}^{n} c_j u_j$ and $X^T u_j = 0$ for any $j \in [n]$, then $X^T u = \sum_{j=1}^{n} c_j X^T u_j = 0$. Contradiction!

Without loss of generality, we suppose that $Au_1 = u_1$ and $X^T u_1 \neq \mathbf{0}_{d_0 \times 1}$. Then we have

$$(U^T\tilde{\Sigma}^{(1)}U)_{11} = \frac{\sigma_w^4}{d_0} \cdot u_1^T \hat{A}XX^T\hat{A}u_1 = \frac{\sigma_w^4}{d_0} \cdot u_1^T XX^T u_1 = \frac{\sigma_w^4}{d_0} \cdot \|X^T u_1\|^2 > 0.$$

It results in

$$\text{tr}(\tilde{\Sigma}^{(L)}) = \text{tr}(U^T\tilde{\Sigma}^{(L)}U) \geq (U^T\tilde{\Sigma}^{(L)}U)_{11} = (\alpha^2 \sigma_w^2 + \beta^2)^{L-1} \cdot \frac{\sigma_w^4}{d_0}\|X^T u_1\|^2. \tag{43}$$

Therefore, if $\alpha^2\sigma_w^2 + \beta^2 > 1$, we have

$$
\begin{aligned}
\lim_{L\to\infty} \mathbf{M}_{\text{FSP}}^{(L)}(\sigma_w^2) &= \lim_{L\to\infty} \mathbb{E}_{H^{(L)}\sim N(\mathbf{0}_n,\tilde{\Sigma}^{(L)})}[\|H^{(L)}\|_{\text{F}}^2/\|X\|_{\text{F}}^2] \\
&= \frac{C}{\|X\|_{\text{F}}^2} \lim_{L\to\infty} \mathbb{E}_{h\sim N(\mathbf{0}_n,\tilde{\Sigma}^{(L)})}[\|h\|^2] \\
&= \frac{C}{\|X\|_F^2} \lim_{L\to\infty} \text{tr}(\tilde{\Sigma}^{(L)}) = +\infty.
\end{aligned}
$$

$\square$

*Proof of part 2 in Theorem 3.3.* For any positive semi-definite matrix $\Sigma \in \mathcal{S}$, we have

$$
\mathbb{E}_{h\sim N(\mathbf{0}_n,\Sigma)}[\text{Dir}(h)] = \mathbb{E}_{h\sim N(\mathbf{0}_n,\Sigma)}[\text{tr}(h^T\hat{L}h)] = \mathbb{E}_{h\sim N(\mathbf{0}_n,\Sigma)}[\text{tr}(\hat{L}hh^T)] = \text{tr}(\hat{L}\Sigma).
$$

So when we want to study $\mathbb{E}_{h\sim N(\mathbf{0}_n,\Sigma)}[\text{Dir}(h)]$, we only need to look at $\text{tr}(\hat{L}\Sigma)$ in the following of the proof.

Since $\hat{A}\hat{L} = \hat{A}(I_n - \hat{A}) = \hat{A} - \hat{A}^2 = (I_n - \hat{A})\hat{A} = \hat{L}\hat{A}$, we multiply $\hat{L}$ on both sides of the second equation in (41) and get

$$
\begin{aligned}
\hat{L}\tilde{\Sigma}^{(l+1)} &= \sigma_w^2\alpha^2 \cdot \hat{L}\hat{A}\tilde{\Sigma}^{(l)}\hat{A} + \beta^2\hat{L}\tilde{\Sigma}^{(l)} \\
&= \sigma_w^2\alpha^2 \cdot \hat{A}\hat{L}\tilde{\Sigma}^{(l)}\hat{A} + \beta^2\hat{L}\tilde{\Sigma}^{(l)}.
\end{aligned}
$$

Then for any $i \in [n]$ and $l \in \mathbb{N}$, we have

$$
\begin{aligned}
(U^T\hat{L}\tilde{\Sigma}^{(l+1)}U)_{ii} &= \sigma_w^2\alpha^2 \cdot \lambda_i(U^T\hat{L}\tilde{\Sigma}^{(l)}U)_{ii}\lambda_i + \beta^2 \cdot (U^T\hat{L}\tilde{\Sigma}^{(l)}U)_{ii} \\
&= (\alpha^2\lambda_i^2\sigma_w^2 + \beta^2) \cdot (U^T\hat{L}\tilde{\Sigma}^{(l)}U)_{ii}.
\end{aligned}
$$

Thus, for any $i \in [n]$ and $L \in \mathbb{N}$, we have

$$
(U^T\hat{L}\tilde{\Sigma}^{(L)}U)_{ii} = (\alpha^2\sigma_w^2\lambda_i^2 + \beta^2)^{L-1} \cdot (U^T\hat{L}\tilde{\Sigma}^{(1)}U)_{ii} \tag{44}
$$

Since $U^T\hat{L}U = U^T(I_n - \hat{A})U = I_n - \Lambda$, we get

$$
U^T\hat{L}\tilde{\Sigma}^{(1)}U = (I_n - \Lambda)U^T\tilde{\Sigma}^{(1)}U
$$

We denote

$$
r_i = (U^T\tilde{\Sigma}^{(1)}U)_{ii}, \quad \text{for any } i \in [n].
$$

Then by (44), we have

$$
(U^T\hat{L}\tilde{\Sigma}^{(L)}U)_{ii} = (\alpha^2\sigma_w^2\lambda_i^2 + \beta^2)^{L-1} \cdot (1 - \lambda_i)r_i,
$$

From Proposition C.1, we have

$$
\begin{aligned}
(U^T\hat{L}\tilde{\Sigma}^{(L)}U)_{ii} &\le (\alpha^2\sigma_w^2\lambda^2 + \beta^2)^L \cdot (1 - \lambda_i)r_i, \quad \text{if } \lambda_i \in (-1,1); \\
(U^T\hat{L}\tilde{\Sigma}^{(L)}U)_{ii} &= 0 = (\alpha^2\sigma_w^2\lambda^2 + \beta^2)^L \cdot (1 - \lambda_i)r_i, \quad \text{if } \lambda_i = 1,
\end{aligned}
$$

where $\lambda = \max_{\lambda_i\neq 1}|\lambda_i| \in [0,1)$. Thus, we get

$$
\text{tr}(\hat{L}\tilde{\Sigma}^{(L)}) = \text{tr}(U^T\hat{L}\tilde{\Sigma}^{(L)}U) \le (\alpha^2\sigma_w^2\lambda^2 + \beta^2)^{L-1} \cdot \sum_{i=1}^n (1 - \lambda_i)r_i.
$$

We conclude that

$$
\mathbb{E}_{H^{(L)}\sim N(\mathbf{0}_n,\tilde{\Sigma}^{(L)})}[\text{Dir}(H^{(L)})] = C \cdot \mathbb{E}_{h\sim N(\mathbf{0}_n,\tilde{\Sigma}^{(L)})}[\text{Dir}(h)] = C \cdot \text{tr}(\hat{L}\tilde{\Sigma}^{(L)})
$$

$$
\le C(\alpha^2\sigma_w^2\lambda^2 + \beta^2)^{L-1} \cdot \sum_{i=1}^n (1 - \lambda_i)r_i.
$$

Then we have

$$
\frac{\mathbb{E}_{H^{(L)}\sim N(\mathbf{0}_n,\tilde{\Sigma}^{(L)})}[\text{Dir}(H^{(L)})]}{(\alpha^2\sigma_w^2 + \beta^2)^{L-1}} \le \left(C\sum_{i=1}^n (1 - \lambda_i)r_i\right) \cdot \left(\frac{\alpha^2\sigma_w^2\lambda^2 + \beta^2}{\alpha^2\sigma_w^2 + \beta^2}\right)^{L-1}.
$$

Since $\alpha^2\sigma_w^2 + \beta^2 > 1$ and $\alpha \neq 0$ as assumed in the statement of this theorem, we have $(\alpha^2\sigma_w^2\lambda^2 + \beta^2)/(\alpha^2\sigma_w^2 + \beta^2) \in [0, 1)$. So we get that

$$\lim_{L \to \infty} \frac{\mathbb{E}_{H^{(L)} \sim N(\mathbf{0}_n, \tilde{\Sigma}^{(L)})}[\text{Dir}(H^{(L)})]}{(\alpha^2\sigma_w^2 + \beta^2)^L} = 0. \tag{45}$$

Recalling (43) in the proof of part 1 for Theorem 3.3, if we define

$$\delta_0 = \frac{\sigma_w^4}{d_0}\|X^T u_1\|^2 \quad \text{and} \quad K = \alpha^2\sigma_w^2 + \beta^2,$$

then given any $L \in \mathbb{N}$, we have

$$\frac{1}{K^{L-1}} \cdot \text{tr}(\tilde{\Sigma}^{(L)}) \geq \delta_0 > 0. \tag{46}$$

Similar to the proof of part 2 in Theorem D.7, we have

$$\mathbb{E}_{H \sim N(\mathbf{0}_n, \tilde{\Sigma}^{(L)})}\left[\frac{\text{Dir}(H)}{\|H\|_{\text{F}}^2}\right]$$

$$= \mathbb{E}_{H \sim N(\mathbf{0}_n, \tilde{\Sigma}^{(L)})}\left[\frac{\text{Dir}(H)}{\|H\|_{\text{F}}^2}\mathbb{1}_{\{\|H\|_{\text{F}}^2 > \epsilon K^{L-1}\}}\right] + \mathbb{E}_{H \sim N(\mathbf{0}_n, \tilde{\Sigma}^{(L)})}\left[\frac{\text{Dir}(H)}{\|H\|_{\text{F}}^2}\mathbb{1}_{\{\|H\|_{\text{F}}^2 \leq \epsilon K^{L-1}\}}\right]$$

$$\leq \frac{\mathbb{E}_{H \sim N(\mathbf{0}_n, \tilde{\Sigma}^{(L)})}[\text{Dir}(H)]}{\epsilon K^{L-1}} + 2 \cdot \mathbb{P}_{H \sim N(\mathbf{0}_n, \tilde{\Sigma}^{(L)})}[\|H\|_{\text{F}}^2 \leq \epsilon K^{L-1}]. \tag{47}$$

For any $L \geq 1$, there exists $i \in [n]$, such that $\tilde{\Sigma}_{ii}^{(L)} \geq \text{tr}(\tilde{\Sigma}^{(L)})/n$. For any $n \times C$ random matrix $H \sim N(\mathbf{0}_n, \tilde{\Sigma}^{(L)})$, it holds that $H_{i,1} \sim N(0, \tilde{\Sigma}_{ii}^{(L)})$. By (46), we have

$$\mathbb{P}_{H \sim N(\mathbf{0}_n, \tilde{\Sigma}^{(L)})}\left[\|H\|_{\text{F}}^2 \leq \epsilon K^{L-1}\right] \leq \mathbb{P}_{H \sim N(\mathbf{0}_n, \tilde{\Sigma}^{(L)})}\left[H_{i,1}^2 \leq \epsilon K^{L-1}\right]$$

$$= \mathbb{P}_{Z \sim N(0,1)}\left[Z^2 \leq \frac{\epsilon K^{L-1}}{\tilde{\Sigma}_{ii}^{(L)}}\right] \leq \mathbb{P}_{Z \sim N(0,1)}\left[Z^2 \leq \frac{\epsilon n K^{L-1}}{\text{tr}(\tilde{\Sigma}^{(L)})}\right] \tag{48}$$

$$\leq \mathbb{P}_{Z \sim N(0,1)}\left[Z^2 \leq \frac{\epsilon n}{\delta_0}\right] = 2\Phi\left(\sqrt{\frac{\epsilon n}{\delta_0}}\right) - 1,$$

where $\Phi(x) = \mathbb{P}_{Z \sim N(0,1)}[Z \leq x]$ denotes the cumulative distribution function of the standard normal distribution $N(0, 1)$.

Combining (47) and (48), we get

$$\mathbb{E}_{H \sim N(\mathbf{0}_n, \tilde{\Sigma}^{(L)})}\left[\frac{\text{Dir}(H)}{\|H\|_{\text{F}}^2}\right] \leq \frac{1}{\epsilon K^{L-1}} \cdot \mathbb{E}_{H \sim N(\mathbf{0}_n, \tilde{\Sigma}^{(L)})}[\text{Dir}(H)] + 4\Phi\left(\sqrt{\frac{\epsilon n}{\delta_0}}\right) - 2.$$

By (45), we let $L \to \infty$ and get

$$\limsup_{L \to \infty} \mathbb{E}_{H \sim N(\mathbf{0}_n, \tilde{\Sigma}^{(L)})}\left[\frac{\text{Dir}(H)}{\|H\|_{\text{F}}^2}\right]$$

$$\leq \frac{1}{\epsilon K^{L-1}} \cdot \limsup_{L \to \infty} \mathbb{E}_{H \sim N(\mathbf{0}_n, \tilde{\Sigma}^{(L)})}[\text{Dir}(H)] + 4\Phi\left(\sqrt{\frac{\epsilon n}{\delta_0}}\right) - 2 \tag{49}$$

$$= \frac{1}{\epsilon} \cdot 0 + 4\Phi\left(\sqrt{\frac{\epsilon n}{\delta_0}}\right) - 2 = 4\Phi\left(\sqrt{\frac{\epsilon n}{\delta_0}}\right) - 2.$$

Notice that the left hand side of (49) is independent of the choice of $\epsilon$. Since $\Phi$ is a continuous map, we let $\epsilon \to 0^+$ and get

$$\limsup_{L \to \infty} \mathbb{E}_{H \sim N(\mathbf{0}_n, \tilde{\Sigma}^{(L)})}\left[\frac{\text{Dir}(H)}{\|H\|_{\text{F}}^2}\right] \leq 4\Phi(0) - 2 = 0.$$

Therefore, we have

$$\lim_{L \to \infty} \mathbf{M}_{\text{GEV}}^{(L)}(\sigma_w^2) = \lim_{L \to \infty} \mathbb{E}_{H \sim N(\mathbf{0}_n, \tilde{\Sigma}^{(L)})}\left[\frac{\text{Dir}(H)}{\|H\|_{\text{F}}^2}\right] = 0.$$

$\square$

# F  BEST REPORTED DEPTHS OF EXISTING OVER-SMOOTHING-RELATED APPROACHES ON OGBN-ARXIV

Table 2: Summary of depths with optimal test accuracies of existing over-smoothing-related approaches on OGNB-Arxiv. The corresponding optimal test accuracies are also indicated in parentheses.

| Models | Depth (test accuracy) |
|---|---|
| GCN (Kipf & Welling, 2017) | 2 (69.53) |
| DropEdge (Rong et al., 2020) | 2 (68.67) |
| PairNorm (Zhao & Akoglu, 2020) | 2 (65.74) |
| DropNode (Huang et al., 2020) | 16 (67.17) |
| MeanNorm (Yang et al., 2020) | 16 (70.40) |
| GroupNorm (Zhou et al., 2020) | 16 (70.50) |
| NodeNorm (Zhou et al., 2021b) | 16 (70.75) |
| GCNII (Chen et al., 2020b) | 16 (72.61) |
| GPRGNN (Chien et al., 2021) | 16 (70.30) |
| DAGNN (Liu et al., 2020) | 16 (71.82) |
| EGNN (Zhou et al., 2021a) | 32 (72.7) |
| JKNet (Xu et al., 2018) | 16 (66.41) |
| APPNP (Gasteiger et al., 2019) | 16 (66.95) |
| SPoG-ResGCN(Ours) | 64 (72.97) |

In Table 2, we summarize the best depths of existing approaches aiming to tackle the over-smoothing issue. The best depth refers to the depth that achieves the highest test accuracy on OGBN-Arxiv. For approaches that utilize random dropping or normalization techniques, we all employ vanilla GCNs as the underlying model. Most of the results are directly cited from Chen et al. (2022b), while the result of EGNN is cited from the original paper (Zhou et al., 2021a). We see that the best performance in most of these studies is still achieved with less than 20 layers, suggesting that the curse of depth continues to constrain the potential of GCNs.

# G  SPoGInit ALGORITHM DETAILS

In Section 4, SPoGInit aims to find a better initialization by minimizing

$$w_1 \mathbf{V}_{\text{FSP}} + w_2 \mathbf{V}_{\text{BSP}} - w_3 \mathbf{M}_{\text{GEV}}^{(L)}.$$

In the implementation of of SPoGInit algorithm, we always use one random weight sample to get point estimates $\hat{\mathbf{V}}_{\text{FSP}}, \hat{\mathbf{V}}_{\text{BSP}}, \hat{\mathbf{M}}_{\text{GEV}}^{(L)}$ of $\mathbf{V}_{\text{FSP}}, \mathbf{V}_{\text{BSP}}, \mathbf{M}_{\text{GEV}}^{(L)}$, respectively. Given any Xavier-initialized $\{\hat{W}^{(l)}\}_{l=1}^L$, SPoGInit scales the weights layer-wise by $\gamma = (\gamma^{(l)})_{l \in [L]} \in \mathbb{R}_{>0}^L$ to yield new initialization $\theta(\gamma) = \{W^{(l)}\}_{l=1}^L = \{\gamma^{(l)} \hat{W}^{(l)}\}_{l=1}^L$ that achieves proper signal propagation. To be more specific, SPoGInit algorithm solves the optimization problem

$$\min_{\gamma} F(\theta(\gamma)) := w_1 \hat{\mathbf{V}}_{\text{FSP}}(\gamma) + w_2 \hat{\mathbf{V}}_{\text{BSP}}(\gamma) - w_3 \hat{\mathbf{M}}_{\text{GEV}}^{(L)}(\gamma), \tag{50}$$

where

$$\hat{\mathbf{V}}_{\text{FSP}} := (\hat{\mathbf{M}}_{\text{FSP}}^{(1)}/\hat{\mathbf{M}}_{\text{FSP}}^{(L-1)} - 1)^2 = \left[ \frac{\|H^{(1)}(\theta(\gamma))\|_{\text{F}}}{\|H^{(L-1)}(\theta(\gamma))\|_{\text{F}}} - 1 \right]^2,$$

$$\hat{\mathbf{V}}_{\text{BSP}} := (\hat{\mathbf{M}}_{\text{BSP}}^{(2)}/\hat{\mathbf{M}}_{\text{BSP}}^{(L-1)} - 1)^2 = \left[ \frac{\|g^{(2)}(\theta(\gamma))\|_{\text{F}}}{\|g^{(L-1)}(\theta(\gamma))\|_{\text{F}}} - 1 \right]^2,$$

$$\hat{\mathbf{M}}_{\text{GEV}}^{(L)} := \frac{\text{Dir}(H^{(L)}(\theta(\gamma))}{\|H^{(L)}(\theta(\gamma)\|_{\text{F}}^2},$$

with $g^{(l)}(\theta(\gamma)) := \partial \ell / \partial W^{(l)}$.

Now we explain SPoGInit (Algorithm 1) in detail.

In lines 2-3, we initialize the weight parameters and weight scales $\gamma^{(l)}(0) = 1$. We iteratively update $\theta(\gamma)$ as follows.

In lines 5-6, at each iteration, we sample random node features from a standard Gaussian distribution and node labels from a discrete uniform distribution. This trick can help enhance data independence and solve over-fitting problems (Dauphin & Schoenholz, 2019).

In line 7, we calculate the objective function $F(\theta(\gamma(t)))$ as defined in (50).

In lines 8-12, we update the weight parameters $\theta(\gamma(t))$ by optimizing the objective function through the projected gradient descent method to the scales $\{\gamma^{(l)}(t)\}_{l=1}^L$ for each layer. We adopt the projected gradient descent method to ensure the scales $\{\gamma^{(l)}(t)\}_{l=1}^L$ remain positive.

---

**Algorithm 1** *SPoGInit*: searching for weight initialization with better Signal Propagation on Graph

1: normalized adjacency matrix $\hat{A}$, input dimension $d_0$, number of labels $C$, network depth $L$, hidden dimension $d$, learning rate $\eta$, total iterations $T$, metric weights $w_1, w_2, w_3$.
2: **initialize** $\gamma^{(l)}(0) = 1$ and sample $\{\hat{W}^{(l)}\}_{l=1}^L$ by Xavier initialization.
3: **initialize** $\theta(\gamma(0)) \triangleq \{W^{(l)}(0)\}_{l=1}^L$ by $W^{(l)}(0) \leftarrow \gamma^{(l)}(0) \cdot \hat{W}^{(l)}$.
4: **for** $t = 0, 1, \cdots, T-1$ **do**
5:      generate input $X(t) \in \mathbb{R}^{n \times d_0}$ with $X(t)_{ik} \overset{\text{iid}}{\sim} N(0,1)$.          ▷ Sampling and Evaluation
6:      generate label $y_i(t) \overset{\text{iid}}{\sim} \text{Uniform}\{1, 2, \ldots, C\}$ for any node $i \in [n]$.
7:      calculate the objective function $F(\theta(\gamma(t)))$ by $\hat{A}, X(t), y(t)$ and $\theta(\gamma(t))$.
8:      **for** layers $l = 1, 2, \ldots, L$ **do**          ▷ Update
9:          $\gamma^{(l)}(t+1) \leftarrow \gamma^{(l)}(t) - \eta \nabla_{\gamma^{(l)}} F(\theta(\gamma(t)))$.
10:          $\gamma^{(l)}(t+1) \leftarrow \text{Proj}_{[10^{-6}, \infty)}(\gamma^{(l)}(t+1))$.
11:          $W^{(l)}(t+1) \leftarrow \gamma^{(l)}(t+1) \cdot \hat{W}^{(l)}$.
12:      $\theta(\gamma(t+1)) \triangleq \{W^{(l)}(t+1)\}_{l=1}^L$.
13: **return** $\theta(\gamma(T))$.

---

Specifically, we explain how to compute the derivative of the objective function with respect to the scale $\gamma$. Through the chain rule, we can calculate the derivative of the objective function with respect

to the scale $\gamma^{(l)}$ as follows:

$$
\begin{aligned}
\frac{\partial F(\theta(\gamma))}{\partial \gamma^{(l)}} &= \sum_{k'=1}^{d_{l-1}} \sum_{k=1}^{d_l} \frac{\partial F(\theta(\gamma))}{\partial W_{k'k}^{(l)}} \frac{\partial W_{k'k}^{(l)}}{\partial \gamma^{(l)}} = \sum_{k'=1}^{d_{l-1}} \sum_{k=1}^{d_l} \frac{\partial F(\theta(\gamma))}{\partial W_{k'k}^{(l)}} \hat{W}_{k'k}^{(l)} \\
&= \sum_{k'=1}^{d_{l-1}} \sum_{k=1}^{d_l} \frac{\partial F(\theta(\gamma))}{\partial W_{k'k}^{(l)}} \frac{W_{k'k}^{(l)}}{\gamma^{(l)}}.
\end{aligned}
$$

In practice, we employ two alternative methods: 1. direct computation as shown above, 2. black-box optimization through forward passes to approximate the gradient, which saves computational resources (Conn et al., 2009).

Additionally, we provide specific hyperparameter choices for SPoGInit in vanilla GCN experiments. We set total iterations $T$ as 500 and learning rate $\eta$ as 0.1. Considering the sensitivity of the training process to weight gradients, we assign a higher weight to the backward propagation. Finally, we set the forward propagation and backward propagation weights as $w_1 = 1$ and $w_2 = 10$. Moreover, to balance the scale of the graph embedding variation, we utilize the inverse of the Dirichlet energy of the input data as the weight: $w_3 = \|X\|_F^2 / \text{Dir}(X)$.

# H SUPPLEMENTAL EXPERIMENT RESULTS

## H.1 DATASETS

**Datasets:** We focus on seven benchmark datasets for semi-supervised node classification. The small-scale datasets include Cora, Pubmed, and Citeseer Yang et al. (2016). The large-scale datasets comprise OGBN-Arxiv, IGB-tiny19, and Arxiv-year. These large-scale datasets are selected from three popular publicly available graph benchmarks: Open Graph Benchmark (OGB) Hu et al. (2020), Illinois Graph Benchmark (IGB) Khatua et al. (2023), and Large Scale Non-Homophilous Graphs Benchmark Lim et al. (2021). We use a standard training/validation/test split Yang et al. (2016) for Cora, Pubmed, and Citeseer datasets. On large-scale datasets, we adopt standard training/validation/test splits. Statistics of the datasets are summarized in Table 3.

Table 3: Statistics of the seven datasets used in the experiments (Section 5 and Appendix H).

| Dataset | Nodes | Features | Edges | Class | Homophily | Training/Validation/Test |
|---------|-------|----------|-------|-------|-----------|--------------------------|
| Cora | 2708 | 1433 | 10556 | 7 | 0.81 | 5.2%/18.5%/36.9% |
| Pubmed | 19717 | 500 | 88648 | 3 | 0.80 | 0.3%/2.5%/5.1% |
| Citeseer | 3327 | 3703 | 9104 | 6 | 0.74 | 3.6%/15.0%/30.1% |
| OGBN-Arxiv | 169343 | 128 | 1166243 | 40 | 0.66 | 53.7%/17.6%/28.7% |
| IGB-Tiny19 | 100000 | 1024 | 447416 | 19 | 0.56 | 60%/20%/20% |
| Arxiv-year | 169343 | 128 | 1166243 | 5 | 0.22 | 50%/25%/25% |

## H.2 EXPERIMENTAL SETTINGS AND HYPERPAMETERS

**Settings for the initialization experiments of vanilla GCN.**

In Figures 4, 8, 9 and 10, the number of hidden units is set to be 64. The models are trained using the Adam optimizer with the tanh activation function. For the Adam optimizer, we set the momentum coefficients to 0.9 and 0.9995, and perform grid searches over the learning rates ranging from $10^{-3}, 10^{-4}, 5 \times 10^{-5}$, to $10^{-5}$. Table 4 reports the settings for training epochs and early stopping patience with different network depths. To investigate the training degradation issue, we exclude dropout Srivastava et al. (2014) and weight decay.

In this work, we replicate all training experiments across three random seeds. Additionally, we replicate all experiments at initialization across 20 random seeds.

Table 4: Epochs settings of Figures 4, 8, and 9.

| GCN layers | Hyperparameters |
|------------|-----------------|
| 4/8/16 layers | epochs; 800, patience: 200 |
| 32 layers | epochs: 1200, patience: 300 |
| 64 layers | epochs: 1500, patience: 375 |
| 128 layers | epochs: 2000, patience: 400 |

**Settings for the experiments of skip-connection-based GCNs.**

- **Model performance.** In Figures 5, 3 (b) and 11, we set the number of hidden units to 64. We add batch normalization and dropout to all models to enhance the generalization performance, as they are commonly used in the training of GNNs on large-scale datasets. The epoch settings of different datasets are as follows: 1000 epochs for OGBN-Arxiv, 1500 epochs for Arxiv-year, and 1000 epochs for IGB-Tiny 19. The Adam optimizer's two momentum coefficients are set to be 0.9 and 0.9995. The weight decay is fixed to 0. We replicate all training experiments across three random seeds. Additional hyperparameters are reported in Table 5.

- **Backward propagation analysis.** For the early training experiments in Figures 7, 14, 15, 16, 17, and the initialization experiments in Figures 12, 13, we maintain most of the settings in the **model performance** experiments (including learning rates, hidden units, initializations, weight decay, and momentum coefficients in Adam). To investigate backward

propagation issues, we exclude batch normalization and dropout. We replicate the early-training experiments across five random seeds.

Table 5: Hyperparameters of Figures 5 and 11

| Models | Hyperparameters |
|---|---|
| JKNet | hidden units: 64, initialization: Xavier, learning rate: 0.005, dropout:0.5. |
| ResGCN | hidden units: 64, initialization: Conventional, learning rate: 0.005, dropout:0.6. |
| GCNII | $\alpha_l$: 0.5, $\lambda$: 0.5, hidden units: 64, initialization: Xavier, learning rate: 0.005, dropout:0.1. |
| SPoG-ResGCN | hidden units: 64, initialization: SPoGInit, learning rate: 0.005, dropout:0.6. |

**Overall settings**

All experiments on large-sized datasets, e.g., OGBN-Arxiv, are conducted on a single NVIDIA V100 32 GB GPU, while small-sized datasets experiments are completed using a single NVIDIA T4 16 GB GPU.

### H.3 ADDITIONAL EXPERIMENTS FOR SPOGINIT

**Signal propagation experiments on additional datasets**

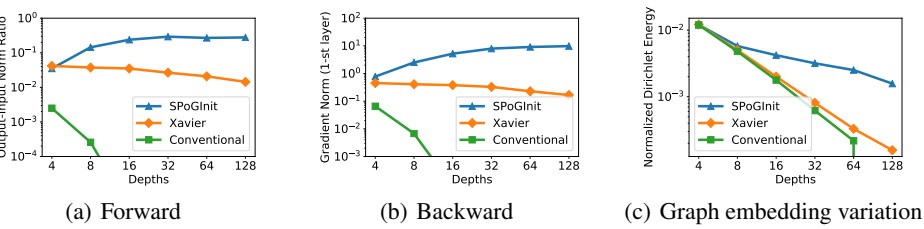

(a) Forward       (b) Backward       (c) Graph embedding variation

Figure 8: The (a) forward metrics, (b) backward metrics, and (c) graph embedding variation metrics of deep vanilla GCNs with different initialization methods on the Citeseer dataset.

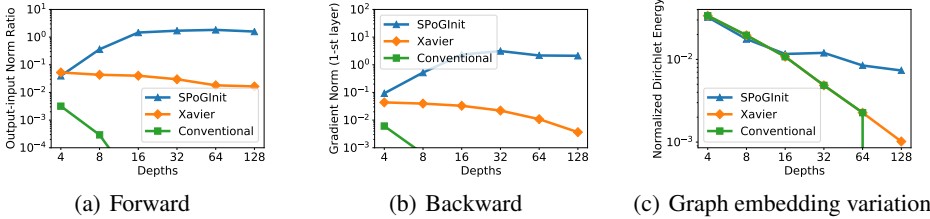

(a) Forward       (b) Backward       (c) Graph embedding variation

Figure 9: The (a) forward metrics, (b) backward metrics, and (c) graph embedding variation metrics of deep vanilla GCNs with different initialization methods on the Pubmed dataset.

In Figures 8 and 9, we present the average forward propagation metrics, backward propagation metrics, and graph embedding variation metrics of tanh-activated vanilla GCNs with various initialization methods, depths, and datasets. We replicate these experiments across 20 random seeds. Results demonstrate that deep vanilla GCNs with Xavier and Conventional initializations suffer from poor forward-backward propagation and graph embedding variation. In contrast, SPoGInit stabilizes the forward-backward propagation and enhances the graph embedding variation.

**Performance on additional datasets**

In Table 6, we present the average training and test accuracies of tanh-activated GCNs with different initialization methods, depths, and datasets. We replicate this experiment across three random seeds. The results show that vanilla GCNs with Xavier initialization and Conventional initialization suffer

Table 6: Training and test accuracies of vanilla GCNs with different initialization methods, depths, and datasets. The abbreviation "OOM" means out of memory.

| Datasets | Init. | Training accuracies for different depths | | | | | | Test accuracies for different depths | | | | | |
|---|---|---|---|---|---|---|---|---|---|---|---|---|---|
| | | 4 | 8 | 16 | 32 | 64 | 128 | 4 | 8 | 16 | 32 | 64 | 128 |
| | Conventional | 100 | 100 | 73.6 | 63.6 | 43.8 | 49.8 | 79.3 | 71.2 | 57.8 | 47.8 | 36.9 | 37.1 |
| Cora | Xavier | 100 | 100 | **100** | 91.0 | 87.4 | 81.0 | 79.4 | **78.4** | 75.2 | 71.6 | 70.5 | 64.8 |
| | SPoGInit | 100 | 100 | 99.3 | **100** | **92.6** | **88.1** | **79.7** | 77.9 | **77.0** | **74.7** | **74.0** | **72.3** |
| | Conventional | 100 | 100 | 88.9 | 73.3 | 75.6 | 60.6 | 76.3 | 72.6 | 67.3 | 68.9 | 62.3 | 49.0 |
| Pubmed | Xavier | 100 | 100 | **100** | 97.8 | **91.7** | 74.4 | **76.6** | 75.9 | 75.9 | **76.3** | **78.1** | 68.7 |
| | SPoGInit | 100 | 100 | 99.4 | **98.3** | 89.4 | **86.1** | 76.3 | **76.4** | **77.9** | 75.9 | 77.2 | **75.9** |
| | Conventional | 100 | 99.2 | 97.8 | 43.1 | 63.6 | 34.2 | 67.6 | 59.3 | 52.1 | 40.2 | 37.8 | 29.3 |
| Citeseer | Xavier | 100 | 100 | 98.1 | 94.7 | 91.9 | 85.6 | 67.5 | **67.5** | 62.3 | 56.5 | 56.7 | 54.1 |
| | SPoGInit | 100 | 100 | **98.3** | 94.7 | **93.6** | **91.4** | **67.8** | 65.1 | 59.9 | **62.2** | **57.7** | **54.9** |
| | Conventional | 74.5 | 70.5 | 50.3 | 31.7 | 27.3 | OOM | 70.1 | 67.8 | 49.9 | 33.2 | 35.9 | OOM |
| OGBN-Arxiv | Xavier | 75.2 | 74.4 | 68.3 | 56.2 | 40.5 | OOM | **70.4** | 68.6 | 66.3 | 57.4 | 39.0 | OOM |
| | SPoGInit | **75.5** | **75.1** | **70.9** | **63.5** | OOM | OOM | 70.3 | **69.2** | **67.7** | **63.4** | OOM | OOM |

from performance degradation on various datasets. Our proposed SPoGInit effectively alleviates the performance degradation and outperforms the baseline initializations in deep GCNs on various datasets.

## H.4 EXPERIMENTS ON MISSING FEATURE FOR SPoGINIT

In order to investigate the performance of GCNs with the SPoGInit method on long-range datasets, we employ the missing feature setting Zhao & Akoglu (2020) to construct the long-range setting. In the missing feature setting, a subset of nodes in the validation and test sets lacks features, and the proportion of these nodes is the missing fraction. In the missing feature setting, with semi-supervised node classification as the fundamental task, the depth of GNNs (number of propagations) plays a crucial role in the endeavor to reconstruct effective feature representations for these nodes.

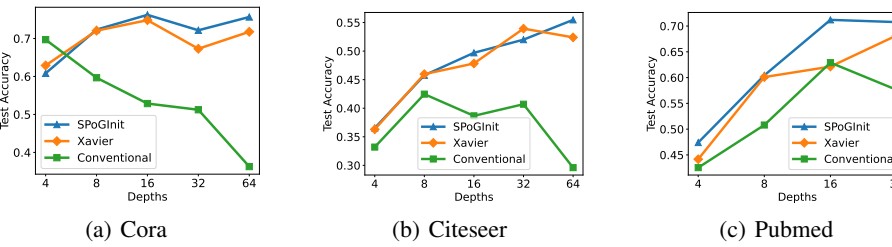

(a) Cora          (b) Citeseer          (c) Pubmed

Figure 10: The test accuracies of tanh-activated deep GCNs with different initializations with $100\%$ missing fraction on (a) the Cora dataset, (b) the Citeseer dataset, and (c) the Pubmed dataset. SPoGInit achieves the best performance at large depth and outperforms the other baselines in this missing feature setting.

In Figure 10, we present the results of GCNs with different initializations and datasets in the missing feature setting. We observe that SPoGInit achieves the best performance at large depths, showing the importance of large depths for capturing long-range information.

## H.5  PERFORMANCE OF SKIP-CONNECTION-BASED GCNS

**Experimental results of tanh-activated models**

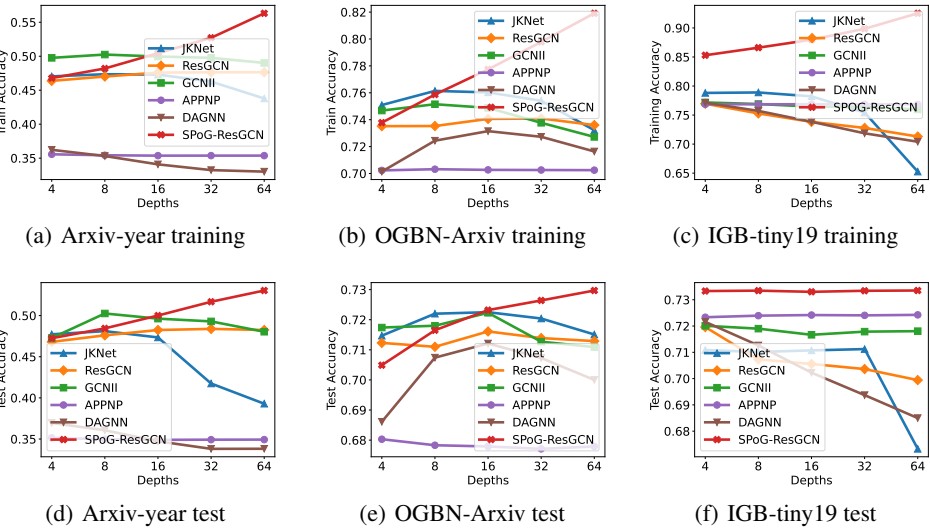

(a) Arxiv-year training     (b) OGBN-Arxiv training     (c) IGB-tiny19 training

(d) Arxiv-year test     (e) OGBN-Arxiv test     (f) IGB-tiny19 test

Figure 11: The average training accuracies (a)-(c) and test accuracies (d)-(f) of different skip-connection-based GCNs with tanh activation on various datasets.

In Figure 11, we present the average training and test accuracies of tanh-activated models with various depths. We see that SPoG-ResGCN stands out by achieving consistent performance gains as the depth increases. Additionally, on the OGBN-Arxiv and Arxiv-year datasets, SPoG-ResGCN achieves around **2.5% gains in test accuracy** by deepening the model from 4 to 64 layers. These results demonstrate that SPoG-ResGCN, with both ReLU and tanh activations, successfully overcomes the curse of depths.

**Optimal performance and the corresponding depths**

In Tables 7 and 8, we present the optimal test accuracies and the corresponding depths for various models and datasets. These values are derived from the results presented in Figures 5 and 11. Notably, we see that SPoG-ResGCN outperforms all the baseline models and achieves optimal performances at the largest depths across most datasets.

Table 7: Optimal test accuracies and the corresponding depths (the numbers in parentheses) of ReLU-activated models.

| Models | OGBN-Arxiv | IGB-Tiny19 | Arxiv-year |
|---|---|---|---|
| APPNP | 68.39±0.13 (4) | 72.41±0.04 (32) | 35.66±0.24 (4) |
| DAGNN | 71.19±0.24 (16) | 72.24±0.05 (4) | 38.29±0.45 (4) |
| JKNet | 72.46±0.17 (16) | 70.96±0.09 (4) | 51.31±0.02 (8) |
| ResGCN | 72.46±0.12 (16) | 72.08±0.15 (4) | 52.98±0.09 (16) |
| GCNII | 72.51±0.39 (8) | 71.96±0.07 (4) | 52.39±0.23 (8) |
| SPoG-ResGCN | **72.76±0.35** (64) | **73.36±0.05** (64) | **54.22±0.14** (64) |

## H.6  BACKWARD METRICS OF BASELINE GCNS WITH SKIP CONNECTIONS

Skip connections significantly change the back-propagation computation. Therefore, in this subsection, we evaluate backward propagation by measuring the gradient norms of four representative layers in an $L$-layer model: the first, the $L/4$-th, the $L/2$-th, and the $3L/4$-th layers. For GCNII, we evaluate its backward propagation metric by the gradient norms of $W_1^{(1)}$ (see Equation 3) in these layers. Additional settings can be seen in Appendix H.2.

Table 8: Optimal test accuracies and the corresponding depths (the numbers in parentheses) of tanh-activated models.

| Models | OGBN-Arxiv | IGB-Tiny19 | Arxiv-year |
|---|---|---|---|
| APPNP | 68.03±0.07 (4) | 72.42±0.12 (16) | 35.14±0.11 (4) |
| DAGNN | 71.21±0.07 (16) | 72.16±0.05 (4) | 36.95±0.05 (4) |
| JKNet | 72.25±0.28 (16) | 71.12±0.03 (32) | 48.11±0.10 (8) |
| ResGCN | 71.61±0.09 (16) | 71.96±0.04 (4) | 48.38±0.06 (32) |
| GCNII | 72.23±0.26 (16) | 72.01±0.12 (4) | 50.25±0.06 (8) |
| SPoG-ResGCN | **72.97±0.16** (64) | **73.35±0.03** (8) | **53.04±0.14** (64) |

**Gradient norms of $\alpha^{(l)}$ in SPoG-ResGCN at initialization**

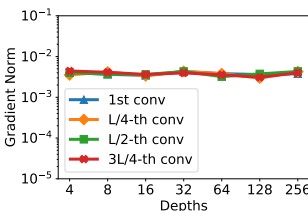

Figure 12: The gradient norms of the $\alpha^{(l)}$ in ReLU-activated SPoG-ResGCNs with various depths and layers at initialization on the Cora dataset. We replicate this experiment across 50 random seeds.

In Figure 12, we present the average gradient norms of $\alpha^{(l)}$ in ReLU-activated SPoG-ResGCNs at initialization. The results demonstrate that $\alpha^{(l)}$ in SPoG-ResGCNs exhibit non-vanishing and stable gradient norms at initialization in various depths and layers.

**Backward metrics of baseline GCNs with skip connections at initialization**

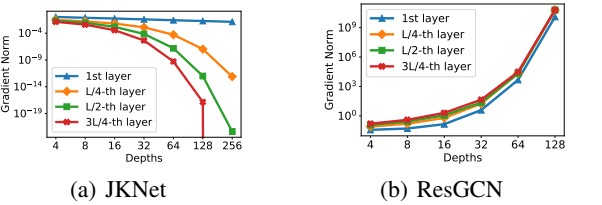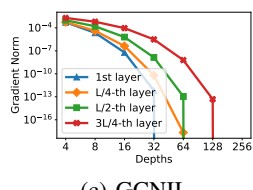

| (a) JKNet | (b) ResGCN | (c) GCNII |
|---|---|---|

Figure 13: The backward metrics of four layers in ReLU-activated baseline models with various depths at initialization on the Cora dataset. We replicate this experiment across 20 random seeds.

In Figure 13, we present the average backward metrics of baselines with different depths at initialization. Results demonstrate that JKNet and GCNII suffer from serious gradient vanishing problems at initialization, while the gradient norms of ResGCN explode at initialization.

**Backward metrics of skip-connections based GCNs during early training**
In Figures 14, 15, 16, and 17, we present the backward metrics of JKNet, ResGCN, GCNII, and SPoG-ResGCN with different depths in early training. We see that, across the early 300 epochs, JKNet exhibits stable backward propagation in its first layer, while the gradient norms of the other layers vanish as the depth increases. Across early 300 epochs, the gradient norms of GCNII vanish as the depths increase, while the gradient norms of ResGCN explode. In contrast, the gradient norms of SPoG-ResGCN quickly improve during early training.

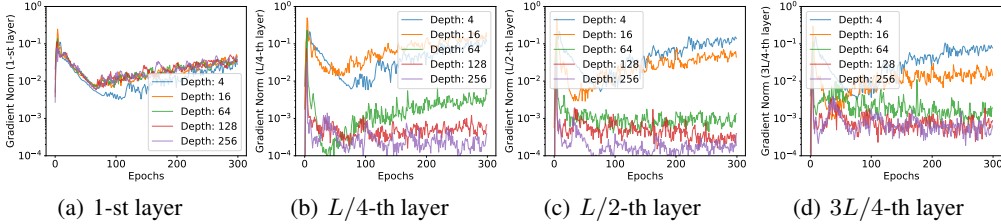

Figure 14: The backward metrics of four layers in ReLU-activated JKNet with different depths in 300 epochs for training on IGB-Tiny19 dataset. We replicate this experiment across five random seeds.

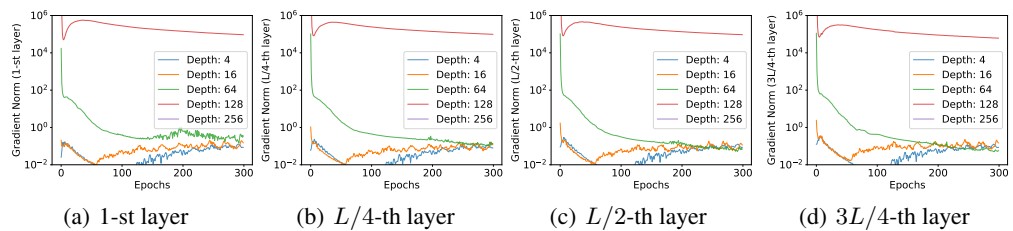

Figure 15: The backward metrics of the four layers in ReLU-activated ResGCN with different depths in 300 epochs for training on IGB-Tiny19 dataset. We replicate this experiment across five random seeds. The disappearing lines are caused by surpassing the machine precision.

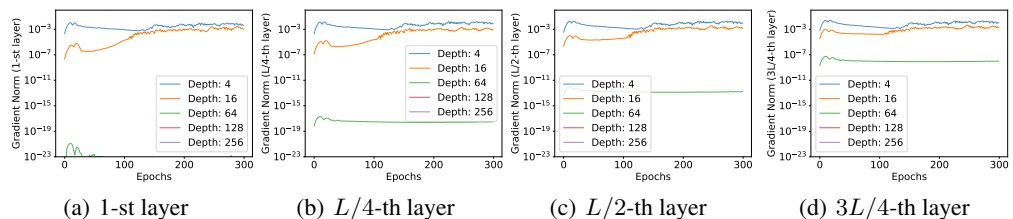

Figure 16: The backward metrics of $W_1^{(l)}$ (see equation (3)) in ReLU-activated GCNII with various depths and layers in 300 epochs for training on IGB-Tiny19 dataset. We replicate this experiment across five random seeds. The disappearing lines are caused by surpassing the machine precision.

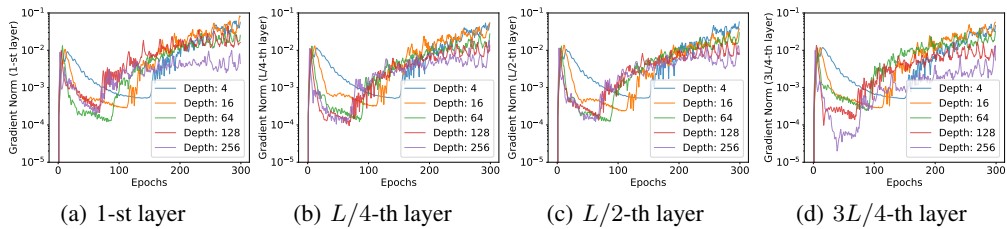

Figure 17: The backward metrics of the four layers of ReLU-activated SPoG-ResGCN with different depths in 300 epochs for training on IGB-Tiny19 dataset. We replicate this experiment across five random seeds.

## I LIMITATION AND NEGATIVE SOCIAL IMPACT

In this paper, we employ signal propagation theory to analyze the curse of depth in Graph Convolutional Networks (GCNs). Additionally, we propose a solution (SPoGInit) to address signal propagation issues and alleviate the curse of depths in GCNs. Interesting directions for future work include applying signal propagation on the GNNs with attention mechanisms. This paper is a theoretical and algorithmic paper on graph neural nets, and does not seem to pose negative social impact.

