# OpenReview forum: "Enhancing Deep Graph Neural Networks via Improving Signal Propagation"
_ICLR.cc/2024/Conference — ICLR 2024 Conference Withdrawn Submission_

### Official Review · Reviewer_S8wQ · 2023-10-31

**Soundness:** 3 good
**Presentation:** 2 fair
**Contribution:** 2 fair
**Rating:** 3
**Confidence:** 4

**Summary:**

Graph convolution networks suffer from the over-smoothing problem as network depth increases, which deteriorates the performance. This paper first illustrates this phenomenon by theoretical and numerical analysis, concluding that with conventional initialization schemes, both vanilla GCN and ResGCN are inclined to over-smoothing in terms of the FSP, BSP and GEV metrics. To tackle this issue, the authors propose SPoGInit, a layer-wise initialization approach aiming to optimize the above metrics. SPoGInit achieves performance enhancements in the node classification task in various datasets.

**Strengths:**

1.	The theoretical analysis part of this paper is solid.

2.	This paper is clear and generally well written.

3.	The performance gains are notable in the experiments.

**Weaknesses:**

1.	The proposed method SPoGInit is essentially pretraining the model on a regularization loss and offers little novelty.

2.	In section 4 the proposed method needs more elucidation, e.g. details on how to apply SPoGInit to a vanilla GCN, and some algorithmic specifics (which are currently in Appendix G).

3.	This paper dedicates a large amount of content to proving the existence of the widely recognized over-smoothing problem, which is less innovative and kind of redundant.

**Questions:**

Same as Weaknesses.

---

### Official Review · Reviewer_c4kx · 2023-10-31

**Soundness:** 3 good
**Presentation:** 3 good
**Contribution:** 2 fair
**Rating:** 5
**Confidence:** 4

**Summary:**

This paper focuses on the performance degradation issue in GNNs. Through the lens of graph signal processing, the authors first analyze why traditional initialization methods hurts deep GNN performance, and then proposes the SpoGInit for training deep GNNs.

**Strengths:**

1. The writing is clear and easy to follow.
2. The authors analyze the performance degradation issue from a new aspect.

**Weaknesses:**

1. Theoretical analysis is not convincing. There are too many assumptions under the theoretical analysis. Sec. 3.1 is based on the assumption of infinite-width limit and Sec. 3. ignores the non-linearity. Thm 3.1 is based on the NNGP correspondence approximation which assumes Gaussian-distributed embeddings. The authors should at least analyze the gap between real GNNs and the assumption. Otherwise, it is hard to say whether the theoretical results make sense or not.
2. The motivation of this paper is unclear: why do we need deep GNNs? According to experimental results on deep GNNs of previous works and also this paper, deep GNNs do not significantly outperform shallow ones on most datasets. In addition, deep GNNs have much larger model sizes and training costs. Then what is the motivation for deepening GNNs? Is it just GNNs are models in deep learning and so they have to be deep?
3. Lack of comparison. Please compare the proposed method with other approaches (i.e., those listed in the related work section) for improving deep GNNs performance.

**Questions:**

Please discuss the relation between this work and ReZero.

---

### Official Review · Reviewer_oZQy · 2023-11-01

**Soundness:** 3 good
**Presentation:** 3 good
**Contribution:** 2 fair
**Rating:** 5
**Confidence:** 4

**Summary:**

This paper studies the curse of depth issue in GNNs from the signal propagation perspective. Specifically, the authors propose three metrics to measure the signal propagation in GNNs, i.e., forward-propagation, backward-propagation, and graph embedding variation (GEV). A new GNN initialization method is designed by minimizing these three metrics. Experimental results demostrate the proposed method can alleviate the oversmoothing issue in GNNs.

**Strengths:**

1. The paper delves both theoretically and empirically into the quality of GNN initialization, focusing on three critical facets.
2. The authors propose the SpogInit method for the initialization of both GCN and residual GCN.
3. Experimental results demonstrate that the proposed method can alleviate the oversmoothing issue in GNNs.

**Weaknesses:**

1. The paper studies the oversmoothing issue in GNNs from the model parameter initialization perspective. However, is the initialization the root cause of the oversmoothing issue in GNNs? Besides, do all three metrics affect the oversmoothing issue? Does well-behaved signal propagation mean better performance?
2. The theoretical analysis is based on the assumption that the hidden dimension is infinite-width. However, the authors only use a fixed hidden dimension during the experiments.
3. What is the time complexity of the proposed initialization method?

**Questions:**

1. The authors claim the oversmoothing issue cannot be resolved by adjusting the weight variance of ReLU-activated GCN. Does the tanh-activated model perform better?
2. The definitions of $V_{FSP}$ and $V_{BSP}$ are different. One uses the first layer, while another uses the second layer. What is the reason?
3. What is the performance of the proposed method with two layers GCN?
4. Can the proposed method be applied to other GNNs, for example, APPNP and GCNII?

---

### Official Review · Reviewer_4Jkm · 2023-11-01

**Soundness:** 2 fair
**Presentation:** 2 fair
**Contribution:** 2 fair
**Rating:** 5
**Confidence:** 4

**Summary:**

Graph Neural Networks (GNNs) deteriorate in performance with increasing depth, known as the "curse of depth." This study analyzes the issue through signal propagation, introducing three metrics for evaluation and proving traditional initialization methods inadequate. To address this problem, the authors introduce a new GCN initialization method called Signal Propagation on Graph (SPoGInit), which effectively mitigates performance degradation in deep GCNs.

**Strengths:**

1. The manuscript is articulated clearly, providing a straightforward narrative that facilitates reader comprehension.

2. The authors have carried out an extensive array of experiments, demonstrating a solid effort to validate their proposed methods.

3. The theoretical analysis is presented with an adequate level of detail, contributing to the overall understanding of the study.

**Weaknesses:**

I have three major concerns about this paper:


1. **Missing Important Baselines**:
   - The authors have limited their comparison to Xavier and Conventional initialization methods, overlooking other significant initialization techniques tailored for graph neural networks. A notable example is Virgo [2], which has been specifically devised for GNNs. To provide a comprehensive evaluation and validate the effectiveness of their proposed method, it is crucial for the authors to include such GNN-specific initialization methods in their experimental comparisons.


2. **Novelty of the Proposed Method**:
   - The proposed SPoGInit method bears a conceptual resemblance to MetaInit, while SPoG-ResGCN shares similarities with ReZero. The authors briefly mention these two prior works in the Appendix, dedicating only a few words to them. Given the substantial similarities, a more thorough and detailed discussion is needed, preferably within the main body of the paper, to critically examine and delineate the differences and advancements made by their methods in comparison to existing ones.
   - The Graph Embedding Variation (GEV) is conceptually identical to concepts proposed in previous works, specifically referenced in [1].
   - The theoretical derivations heavily rely on frameworks previously applied to conventional neural networks. While the findings are applicable to GCNs, they do not significantly deviate from those associated with conventional neural networks. Considering that GCNs inherently deal with graph-structured data, an initialization method that more directly reflects and leverages the unique properties of graphs is expected.

3. **Improper Evaluation**:
   - The evaluation predominantly relies on the ogbn-arxiv dataset, where the authors claim a 2.2% improvement in test accuracy with increasing network depth. However, the homophilous nature of ogbn-arxiv means it doesn’t significantly benefit from deeper architectures. In fact, state-of-the-art performance on this dataset surpasses that achieved by SPoGInit, and interestingly, these results were obtained using shallower GNNs. The heavy focus on this particular dataset in the experimental evaluations does not adequately justify the need to address the 'curse of depth' and is insufficient to validate the effectiveness of SPoGInit across diverse graph structures and datasets.


[1] Samuel S. Schoenholz, Justin Gilmer, Surya Ganguli, and Jascha Sohl-Dickstein. "Deep information propagation." In International Conference on Learning Representations, 2017.

[2] Li, Jiahang, Yakun Song, Xiang Song, and David Wipf. "On the initialization of graph neural networks." In International Conference on Machine Learning,  2023.

**Questions:**

1. Could you provide a comparison of your proposed methods with Virgo, a notable initialization technique specifically designed for graph neural networks?
2. Could you evaluate the performance of your methods on datasets that inherently require deeper network architectures?